# Worldwide divergence of values

Joshua Conrad Jackson [1] ✉ & Danila Medvedev [1] ✉

Social scientists have long debated the nature of cultural change in a modernizing and globalizing world. Some scholars predicted that national cultures would converge by adopting social values typical of Western democracies. Others predicted that cultural differences in values would persist or even increase over time. We test these competing predictions by analyzing survey data from 1981 to 2022 ($n = 406{,}185$) from 76 national cultures. We find evidence of global value divergence. Values emphasizing tolerance and self-expression have diverged most sharply, especially between high-income Western countries and the rest of the world. We also find that countries with similar per-capita GDP levels have held similar values over the last 40 years. Over time, however, geographic proximity has emerged as an increasingly strong correlate of value similarity, indicating that values have diverged globally but converged regionally.

Cultural groups vary not only in their customs and traditions, but also in their values. Different national cultures place different degrees of emphasis on the collective vs. the individual[1], openness vs. obedience[2,3], and faith vs. skepticism[4]. Understanding this variation has become a central goal in the scientific study of culture, since social values shape international conflict, economic climate, and law. Studies seek to identify historical origins of cultural differences in values[5–7] and estimate how values have changed across cultures in modern history[8,9]. A fundamental question for this research is whether social values are converging or diverging across cultures. Globalization and imperialism have homogenized many forms of culture, including language[10] and religion[11]. Have values also converged across cultures in recent history, or have they remained an area of persistent or even increasing cultural divergence?

Traditional modernization theories famously predicted a convergence in social values. Inspired by the philosophies of Marx and Hegel, modernization theorists suggested that the end of the Cold War and the rise of globalization would catalyze the worldwide spread of a "universal civilization" with liberal and individualizing values that emphasize the primacy of personal rights and freedoms[12–15]. Others suggested that the global diffusion of industrialization would break down traditional culture and replace it with "modern" class structures and values[16,17]. These perspectives shared an assumption of unilineal modernization: that modern technology and globalization should lead world cultures to increasingly resemble democratic Western nations.

Scholars became more skeptical of these theories as cultural conflicts emerged across the globe throughout the 21st century.

New theoretical models arose during this time, with varying levels of overlap with the older theories. Inglehart's postmaterialist thesis suggested that globalization alone does not lead to cultural convergence, but that economic development in particular induces a shift from values prioritizing group obedience to values prioritizing self-expression and individual autonomy[9,18]. Welzel labeled these latter values as "emancipative," and proposed a "human empowerment" sequence in which wealth and security encourage cultures to espouse more emancipatory values, which in turn foster participatory democracy[8]. Emancipative values bear a conceptual overlap and statistical correlation with individualizing values. Both emphasize the autonomy and needs of the individual over those of the group[8]. The post-materialist thesis therefore reproduces the assumption that cultural convergence will inherently be in the direction of Western individualism, but adds the caveat that this convergence requires economic prosperity and financial security.

Other theories broke further from modernization theories. Eisenstadt's multiple modernities thesis proposed that economic development would not be a Westernizing force, but would set national cultures on unique paths towards modernization[19,20]. Tomlinson similarly argued that globalization might promote states in the developing world to actively develop and reinforce distinct national identities rather than Westernize[21]. Huntington went one step further, predicting that post-Cold War globalization would lead to a resurgence of cultural divides based on religious and linguistic differences associated with historical civilizations[22].

[1]Booth School of Business, University of Chicago, Chicago, USA. ✉e-mail: joshua.jackson@chicagobooth.edu; dmedvede@chicagobooth.edu

Together, these theories offer a spectrum of competing hypotheses. From one perspective, high- and low-income countries should be experiencing a gradual convergence of values brought on by globalization. From another perspective, this convergence might only characterize countries that have become wealthier over time. And from the other extreme, countries should be diverging in their values, and this divergence might even be sharpest among high-income countries.

Evolutionary models also offer mixed predictions for value convergence. These models state that cultural differences can arise from socioecological pressures involving subsistence style[23], population pressure[7], resource scarcity[2], climate[24], and pathogen load[25,26]. Cultural values and norms often promote behavior that is adaptive in light of these pressures[27]. Sometimes these values and norms emerge over long periods of history, but they also can change quickly[25,26], which evolutionary psychologists have termed "evoked culture"[28,29].

Given the co-evolution between culture and ecology, one might expect that values should converge if people's environments have also become more similar. In some specific ways, environments do seem to be more similar. The decline of biodiversity and the diffusion of new technology mean that people around the world consume the same foods and use the same products with greater ease than ever before[21,30]. However, because socioecological diversity is multidimensional and hard-to-quantify, the question of whether there has been a definitive trend toward environmental homogeneity remains open. Even if there is a trend towards environmental homogeneity, it could be affecting countries in the same regions more than countries across the globe[31]. Studies have found that countries in the same trading blocs have developed more similar economic, demographic, financial, and political characteristics over time, whereas countries from different trading blocs have not become more similar[32].

The influence of Western mass media may be the most intuitive force of cultural convergence. Social learning is the dominant method through which humans transmit cultural values and norms[33], and the diffusion of films, television, the internet, and educational materials have made it easier to learn about the United States that any other country[34]. A recent analysis found that educational attainment correlated with cultural similarity to the United States across the world, suggesting educational attainment may propagate Western values[35]. Yet there is less evidence that non-educational mass media is a Westernizing force. Some countries specifically ban or regulate Western media[36,37]. Even when foreign media is unregulated, people often prefer national and regional content[38]. And when people do consume foreign media, there is no clear evidence that it leads them to accept foreign values. Studies that use media as an intervention tool to change norms in non-Western cultural groups have been careful to work with local organizations to produce culture-specific productions instead of emulating Western media[39,40]. A mass media perspective, like an ecological perspective, offers no clear predictions for whether national cultures are converging or diverging in their values.

The World Values Survey (WVS) has become the proving ground for hypotheses about contemporary value change[41]. The WVS is a multi-panel survey of 450,869 demographically representative people across 105 countries, with multiple waves of data from 76 of these countries (406,185 people). The first timepoint (wave) of this survey took place in 1981, and the seventh timepoint completed data collection in 2021. Few studies have examined every WVS timepoint, but many have analyzed changes in the mean level of key values or value dimensions across subsets of timepoint and countries[32,42–45]. Some studies have argued for global trends on specific values. One analysis documented a global rise of individualism[46]. Another study reported that emancipative values are diffusing around the world, but this diffusion has been more rapid in liberal democracies than in other government types[47]. Other research has focused on specific world regions. For example, membership within the European Union (E.U.) is associated with greater value similarity with other E.U. countries over time but also with value divergence from Central Asian countries[45,48]. The most thorough analysis to date compared responses on three WVS questionnaires across 18 countries at two timepoints (1990s vs. 2010s)[49]. It reported a worldwide shift among these countries towards cultural traits typical of rich Western individualist countries, but also heterogeneity in effect sizes across the questionnaires. Together, these studies provide insights into contemporary changes in specific social values or among small samples of countries but few conclusions about worldwide trends towards value convergence vs. divergence across a large and heterogeneous pool of countries.

In this work, we develop a general method to test whether social values have diverged or converged across the 76 countries that have provided multiple waves of WVS data. Instead of focusing on the mean levels of particular values over time, as in most prior studies, we create metrics that explicitly measure variation in values (see Table 1). These measures are meant to represent the same outcome—convergence vs. divergence of values across world nations. However, they vary in their level of analysis. The first measure, "value variation," focuses on divergence at the level of the WVS item whereas the second measure, "value distinctiveness," focuses on divergence at the country level. Table 1 describes both measures and defines a third measure called

**Tables 1 | Descriptions of key measures**

| Measure | Unit of Analysis | Methodology | Interpretation |
|---|---|---|---|
| Value Variation | Item | We normalize responses to the 40 social value items that have been included in each WVS timepoint to a 0–1 scale. We then compute the standard deviation for the global distribution of country means at each timepoint. Higher standard deviations represent more value variation. We can calculate this trend for a single item, or across all items. | A rise in item-level value variation over time indicates value divergence, whereas a decline indicates convergence. |
| Value Distinctiveness | Country | After normalizing item responses and calculating country means for each item, we take the median value of these country means. This represents the global average of a given item at a particular point in time. We then take the absolute difference between each country mean and the global average. Higher values of this absolute difference mean that a country's values are dissimilar than most other countries. | Countries with high value distinctiveness across all values are dissimilar from countries in the rest of the world. If value distinctiveness increases in a country over time, this suggests that values in that country are diverging from those of other countries. If value distinctiveness is rising across all countries, this indicates that countries are becoming more dissimilar from each other over time, indicating global value divergence. |
| Within-Country Heterogeneity | Country | We calculated within-country heterogeneity by applying the same procedure for estimating between-nation value variation to estimate variation of values across people living in the *same* nation. | Within-country heterogeneity measures whether individuals within countries hold different values. For example, high within-country heterogeneity in the United States might arise because liberal and conservative Americans hold very different values. |

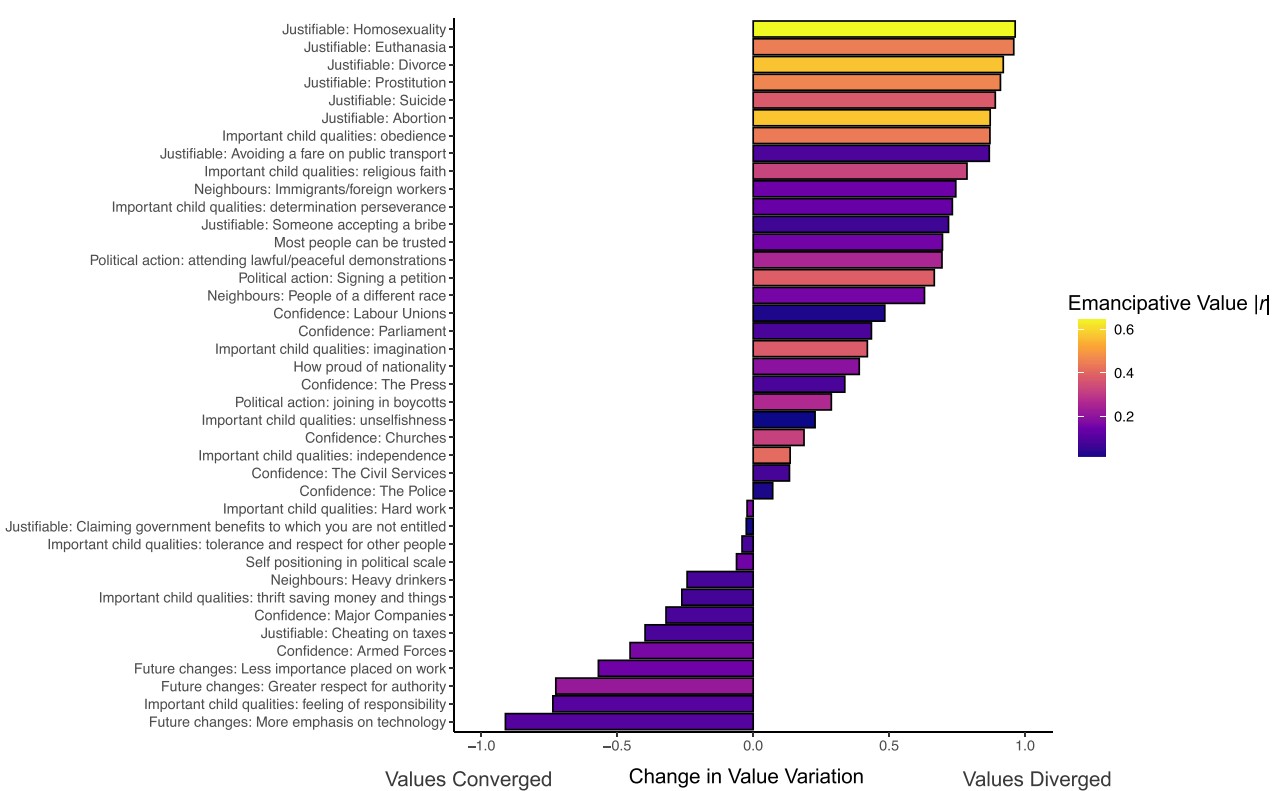

**Fig. 1 | Change in global value variation (*SD* of normalized country means) for each of the 40 WVS items in our analysis.** Each bar represents a correlation coefficient between timepoint and value variation for a given value (value labels are listed on the y-axis). Positive bars indicate that value variation is rising, and that values are diverging. Negative bars indicate that value variation is falling, and values are converging. The fill color indicates the absolute value of the correlation of each item with Welzel's index of emancipative values[8]; brighter bars correlate either highly positively or negatively on this index.

"within-country heterogeneity," which we use in secondary analyses to represent the homogeneity vs. heterogeneity of values within a nation. We define all three measures here so that readers can distinguish between them in our presentation of results.

Our main analyses test whether value variation and value distinctiveness are increasing or decreasing over time. Increases indicate value divergence, and decreases indicate value convergence. We try to test this relationship as rigorously as possible. For example, we include several robustness checks to ensure that the results are not driven by the way we normalize the items or the sampling strategy of the WVS. In our main text, we complement our primary analyses by analyzing different subsets of countries and by exploring geopolitical variables that can explain country-level clustering based on value similarity. Supplemental analyses show that results replicate with different approaches to normalizing means for items that have different response scales and across demographically weighted and unweighted country-level scores. Our materials and methods summarize our analysis procedure, and the Supplementary Methods have an "extended" materials and methods section with more detail.

Here we use this approach to show that values have diverged across national cultures over time. This value divergence mainly characterizes a growing gap between high-income Western countries and the rest of the world on emancipative values. We also find that worldwide value divergence has been accompanied by value convergence among countries in the same region.

## Results

### Value divergence at the item level

We first examined general trends towards value convergence or divergence using our value variation measure, which allowed us to estimate effects at the item level. Our results strongly supported value

divergence. A mixed effects model with value variation nested in the 40 items found that timepoint has been significantly associated with greater value variation, $b = 0.004$, $SE = 0.0007$, $t(239) = 5.12$, $p < 0.001$, $\beta = 0.17$, *95% CIs* [0.002, 0.005]. We replicated this result using a different approach in which we correlated timepoint with value variation separately for each of the 40 value items. Of the 40 values, we found that 27 have diverged over time, with a positive median correlation of 0.28 between timepoint and value variation, $t(39) = 3.30$, $p = 0.002$, $M_{diff} = 0.28$, *95% CIs* [0.11, 0.45]. Coefficients associated with each item are displayed in Fig. 1.

Have certain kinds of values diverged more than others? To test this question, we measured how much each item related to Welzel's dimensions of "sacred vs. secular" values and "emancipative vs. obedient" values[8,27]. Welzel developed these dimensions to differentiate between values that uphold or reject tradition and religion (sacred-secular values) from those that foster or restrict the freedom of the individual from the group (emancipative-obedient values)[8]. We found that a broad set of items loaded on the secular-sacred dimension, ranging from the justifiability of cheating on taxes (0.46), to the confidence in churches (0.46), to the justifiability of euthanasia (0.37). A narrower set of items loaded on the emancipative-obedient dimension. Of the examples above, only justifiability of euthanasia loaded above 0.35 (0.44).

We found that the rate of value divergence correlated with loading on the emancipative-obedient dimension, $r(38) = 0.54$, $p < 0.001$, but not the sacred-secular dimension, $r(38) = 0.19$, $p = 0.237$. This is clearly visible in Fig. 1. The 7 items with the highest divergence scores each have high loadings on the emancipative-obedient dimension. These values were (1) justifiability of homosexuality, (2) justifiability of euthanasia, (3) importance of obedience of children, (4) justifiability of divorce, (5) justifiability of prostitution, (6) justifiability of suicide, and

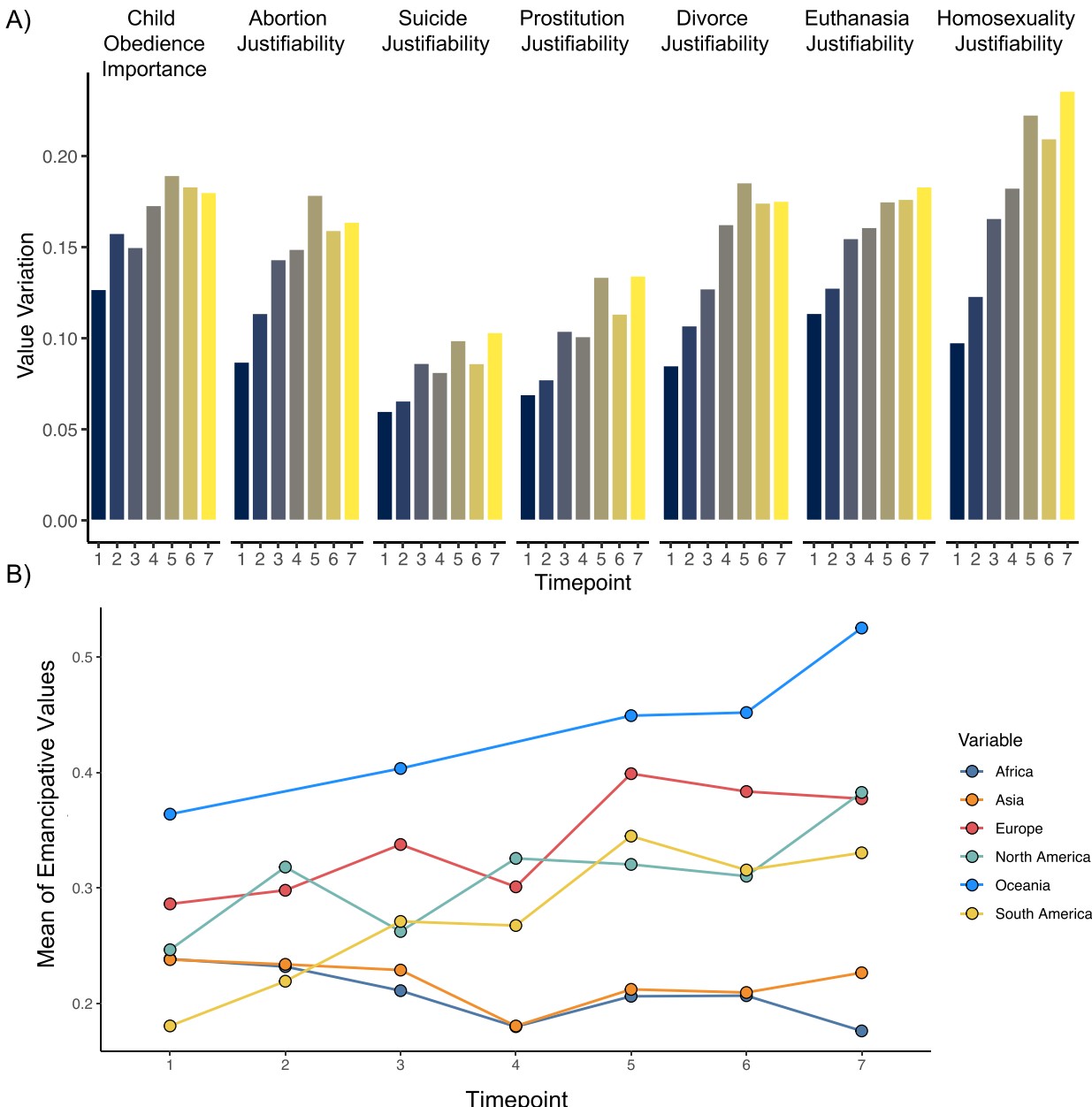

**Fig. 2 | Divergence of key values over time. A** Value variation (*SD* of the global distribution of normalized country means) at each WVS timepoint for the 7 items which have diverged most over time. Item labels are at the top of the figure. Bars are shaded by timepoint. **B** Normalized mean endorsement of the same 7 items by timepoint, with separate lines for continents. All items are coded so that higher scores reflect higher loadings on the index of emancipative values.

(7) justifiability of abortion. National cultures are diverging the most on their tolerance of individual expression versus emphasis on group obedience. Supplementary Fig. 1 reproduces Fig. 1 with shading based on secular-sacred instead of emancipative-obedient loading.

Our approach allowed us to understand the nature and magnitude of divergence on emancipative values. From the first to the last timepoint, value variation across countries increased 141% for justifiability of homosexuality, 94% for justifiability of prostitution, 61% for justifiability of euthanasia, and 42% for importance of childhood obedience (Fig. 2).

Divergence on emancipative values is increasingly distinguishing Western countries from non-Western ones. Consider the case of Australia and Pakistan. The first time Australians were surveyed in the WVS, 39% of participants cited childhood obedience as an important quality in children, and participants rated divorce as more unjustifiable than justifiable (0.45). When Pakistanis were first surveyed, their responses

were not so different: 32% cited childhood obedience as an important childhood quality and participants rated divorce as more unjustifiable than justifiable (0.10). Over time, however, these views diverged. The last time that they were surveyed, only 18% of Australians compared to 49% of Pakistanis cited childhood obedience as an important quality, and Australians viewed divorce as much more justifiable (0.74) than Pakistanis (0.15). From the 1980s to the 2020s, similar fault lines emerged between Western and non-Western countries.

Rises in value variation for the 7 most divergent items are displayed in Fig. 2A. The clearest rises in value variation come from timepoints 1–5 and plateau across timepoints 5–7. Figure 2B illustrates changes over time in the means of these 7 values, which are aggregated to the continent level and coded such that higher values mean more emancipative values. This plot shows that Oceanic, European, North American, and South American countries have progressively endorsed more emancipative values, whereas endorsement of these values has

**Table 2 | Associations with value distinctiveness in multiple regression**

|  | *b* (SE) | *t*-value | df | *p*-value | 95% CIs |
|---|---|---|---|---|---|
| **Model 1** | | | | | |
| Timepoint (C) | 0.002 (0.003) | 0.67 | 49.82 | 0.503 | −0.004, 0.009 |
| Timepoint (L) | 0.003 (0.001) | 4.92 | 10517.38 | <0.001 | 0.002, 0.004 |
| **Model 2** | | | | | |
| Timepoint (C) | −0.003 (0.003) | −0.76 | 77.15 | 0.450 | −0.01, 0.005 |
| Timepoint (L) | −0.001 (0.001) | −0.94 | 415.31 | 0.347 | −0.003, 0.001 |
| GDP Per Capita | 0.08 (0.01) | 8.05 | 600.59 | <0.001 | 0.06, 0.10 |
| Gini | 0.03 (0.02) | 1.68 | 223.21 | 0.094 | −0.005, 0.06 |
| Globalization | 0.004 (0.02) | 0.23 | 541.20 | 0.822 | −0.03, 0.03 |
| Political Rights | −0.01 (0.01) | −1.89 | 2114.83 | 0.059 | −0.02, 0.0004 |
| Distance from Equator | −0.01 (0.01) | −0.91 | 65.01 | 0.367 | −0.03, 0.01 |

Subscript (C) denotes cohort effects (due to changes in the WVS sample composition). Subscript (L) denotes longitudinal effects (due to changes in the countries over time). For all tables reporting inferential analyses in the main text and supplementary materials, statistical tests are two-sided and there are no corrections for multiple comparisons.

been stable across Asian and African countries. Divergence on some values is driven by countries moving in opposite directions, as in the case of Australia vs. Pakistan on the value of childhood obedience (see Supplementary Table 16 for other examples). Other values diverged because their mean rate of endorsement changed in some countries but remained the same in others. We provide detailed methodological details about these items and countries in our Supplementary Information (e.g., Supplementary Tables 2–5).

### Value divergence at the country level

We next replicated and extended these findings using our country-level value distinctiveness measure, in which we calculated each country's deviation from the global median of each value. We first regressed this distinctiveness score on timepoint in a mixed effects model with observations nested in values, countries, and continents. This approach helped us account for the non-independence of countries within the same continent in our data analysis. Value distinctiveness has been rising over time, $b = 0.003$, $SE = 0.0005$, $t(9290) = 4.98$, $p < 0.001$, $\beta = 0.05$, 95% CIs [0.002, 0.004], indicating that countries have diverged in their values. We replicated this effect while controlling for spatial autocorrelation more continuously using an approach similar to the one we used to calculate value distinctiveness, $b = 0.003$, $SE = 0.0005$, $t(9351) = 5.18$, $p < 0.001$, $\beta = 0.05$, 95% CIs [0.002, 0.004] (see Supplementary Methods for details). This continuous method further addressed the concern that our results might have been biased by interdependence of datapoints, often called "Galton's problem."

Another concern is that these findings might be confounded with cohort effects, meaning that value divergence has resulted in changes in the WVS sampling strategy over time. If the cohorts of the WVS are becoming increasingly diverse, then the mean level of value distinctiveness would rise even if countries' values stayed consistent over time. We took several steps to address this concern. First, we used centering to separate each country's general value distinctiveness across all waves (representing a cohort effect) from its change over time from wave to wave (representing a longitudinal effect). A model including both variables found that the longitudinal effect was significant, $b = 0.003$, $SE = 0.0006$, $t(10,520) = 4.92$, $p < 0.001$, $\beta = 0.04$, 95% CIs [0.002, 0.004], but the cohort effect was not, $b = 0.002$, $SE = 0.003$, $t(49.82) = 0.68$, $p = 0.503$, $\beta = 0.02$, 95% CIs [−0.004, 0.009].

Second, we recomputed value variation and replicated our models among subsamples of countries that had less turnover and hence less susceptibility to cohort changes. We replicated value divergence among the 54 countries that participated in at least 3 waves, $b = 0.003$, $SE = 0.0006$, $t(8599) = 4.82$, $p < 0.001$, $\beta = 0.05$, 95% CIs [0.002, 0.004], the 32 countries that participated in at least 4 waves, $b = 0.003$, $SE = 0.0006$, $t(6145) = 4.12$, $p < 0.001$, $\beta = 0.05$, 95% CIs [0.001, 0.004], and the 18 countries that participated in at least 5 waves, $b = 0.003$,

$SE = 0.0007$, $t(4059) = 3.73$, $p < 0.001$, $\beta = 0.05$, 95% CIs [0.001, 0.004]. These results provided further evidence that value divergence is not an artifact of changing sampling strategy over time, but arose from national cultures changing in diverging directions over time.

Third, we replicated the finding in sample with no turnover in sample composition—a subset of 33 countries that provided data in the 1990s, 2000s, and 2010s, which were the three decades with the greatest WVS coverage. Value distinctiveness was higher in the 2000s, $b = 0.01$, $SE = 0.003$, $t(78) = 4.05$, $p < 0.001$, $\beta = 0.29$, 95% CIs [0.007, 0.02], and the 2010s, $b = 0.01$, $SE = 0.003$, $t(78) = 2.66$, $p = 0.010$, $\beta = 0.19$, 95% CIs [0.002, 0.02], compared to the 1990s. However, we found no significant difference between the 2000s vs. 2010s, $b = -0.004$, $SE = 0.003$, $t(39) = -1.41$, $p = 0.167$, $\beta = -0.10$, 95% CIs [−0.01, 0.002]. This decade-based analysis provides evidence for value divergence even when we fix the country sample over time. It also suggests that value divergence has been non-linear. In our Supplementary Methods, we evaluate different non-linear models of value divergence. These models suggest that the pace of value divergence has gradually slowed over time, rather than halting at a particular point.

Analyzing value distinctiveness across countries also allowed us to estimate which countries have become most dissimilar from the rest of the world. Inglehart's post-materialist thesis suggests that high-income countries may hold especially distinctive values, at least in the domain of morality and tolerance. However, a range of other geopolitical variables could also lead cultures to adopt distinctive social values, including wealth inequality[50], distance from the equator[51], globalization[19], and the presence of a liberal democracy[47]. We accessed data on these geopolitical metrics over time, and tested whether they could predict which countries were more distinctive and which countries were less distinctive. We retained the random effects structure from our previous mixed effects models to fit these results without violating any model assumptions.

In our regression analysis, GDP per capita was the only variable that significantly predicted value distinctiveness (see Table 2), b = 0.08, $SE = 0.01$, $t(600.59) = 8.05$, $p < 0.001$, $\beta = 0.18$, 95% CIs [0.06, 0.10], with the positive coefficient indicating that higher-income countries have more distinctive values than lower-income countries. No other predictors reached significance ($p$s > 0.05). In our Supplementary Table 8, we show that other geopolitical variables are not significantly linked to value distinctiveness, even when we remove GDP per capita from the model.

Further analyses found that the association between GDP per capita and value distinctiveness varied across world region. Wealth was associated with value distinctiveness among European countries, $b = 0.08$, $SE = 0.02$, $t(18.76) = 3.63$, $p = 0.002$, $\beta = 0.18$, 95% CIs [0.04, 0.12]. However, in a regression model where we interacted GDP per

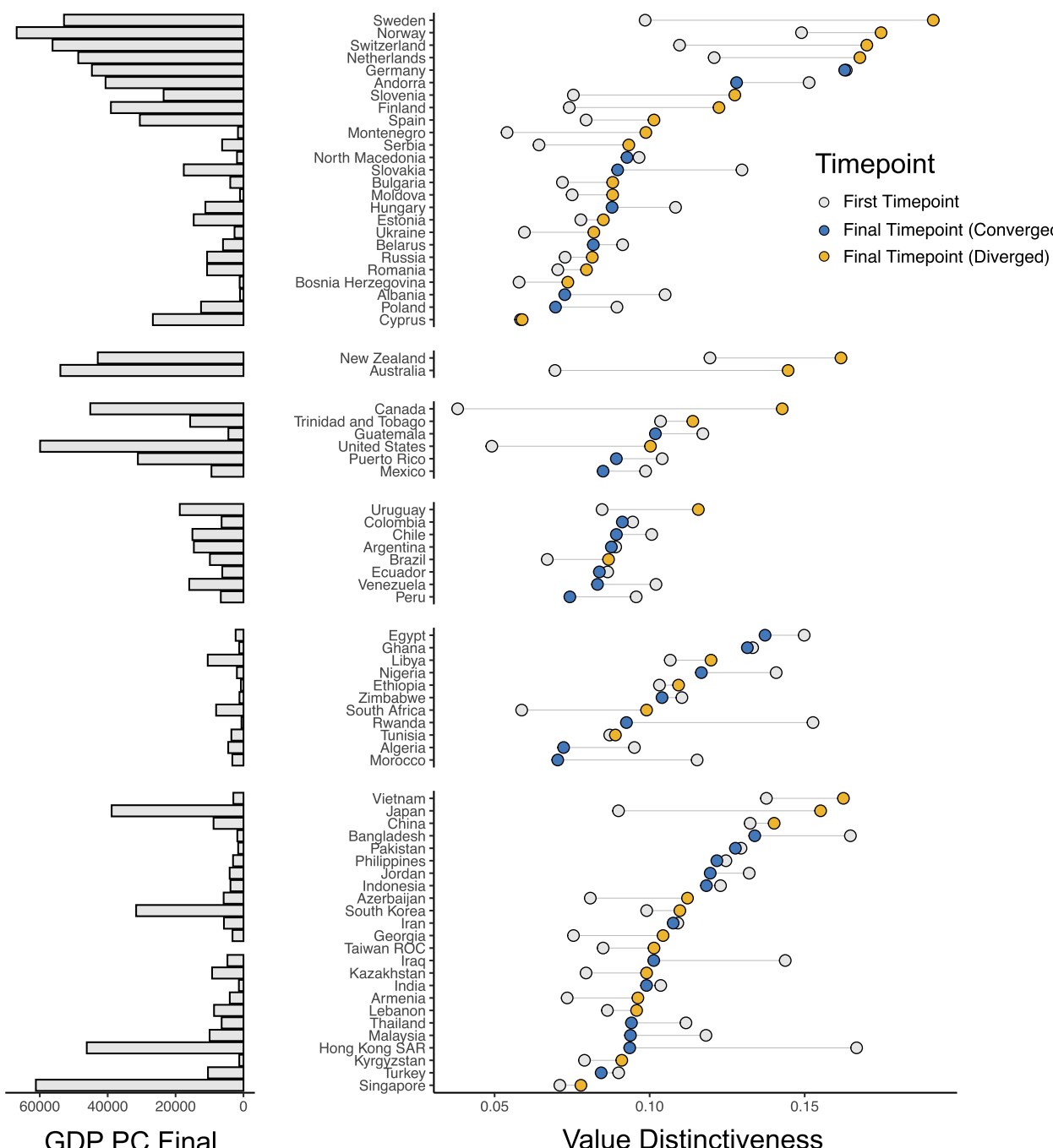

**Fig. 3 | Value distinctiveness of each country over time.** Value distinctiveness is the distance between values in a given country and the global median. Countries are organized by continent, and within continent, by value distinctiveness at the final timepoint in which the country reported data. Yellow (blue) nodes represent countries whose distinctiveness scores have increased (decreased) over time. Gray nodes represent value distinctiveness at the first timepoint. The GDP Per Capita of each country (measured at the final timepoint for each country) is displayed to the left of the plot.

capita with continent dummy-codes, we found that, compared to Europe, the effect of GDP per capita was significantly weaker in Asia, $b = -0.10$, $SE = 0.04$, $t(16.66) = -2.84$, $p = 0.011$, $\beta = -0.23$, 95% CIs [−0.16, −0.04], and Africa, $b = -0.21$, $SE = 0.10$, $t(122.60) = -2.13$, $p = 0.036$, $\beta = -0.50$, 95% CIs [−0.40, −0.02]. The association between GDP per capita and value distinctiveness was non-significant in both Africa, $b = -0.14$, $SE = 0.10$, $t(109.20) = -1.38$, $p = 0.171$, $\beta = -0.32$, 95% CIs [−0.32, 0.06], and Asia, $b = -0.02$, $SE = 0.03$, $t(16.24) = -0.78$, $p = 0.448$, $\beta = -0.05$, 95% CIs [−0.08, 0.03]. These continent comparison models were based on smaller subsets of countries, with 48 countries in the Europe vs. Asia comparison (simulated power =

80.20%) and 36 countries in the Europe vs. Africa comparison (simulated power = 98.00%). Readers should therefore interpret these findings with more caution than the tests that included the full sample of countries. Supplementary Table 13 summarizes the complete set of coefficients for these models. Our Supplementary Methods provide more details about the power analysis simulations.

Figure 3 illustrates many of the findings that we have presented in a single plot, with countries ordered along the y-axis based on their continent and on their level of value distinctiveness within continents. The x-axis illustrates each country's value distinctiveness score at the first WVS timepoint and the final WVS timepoint, and each country's

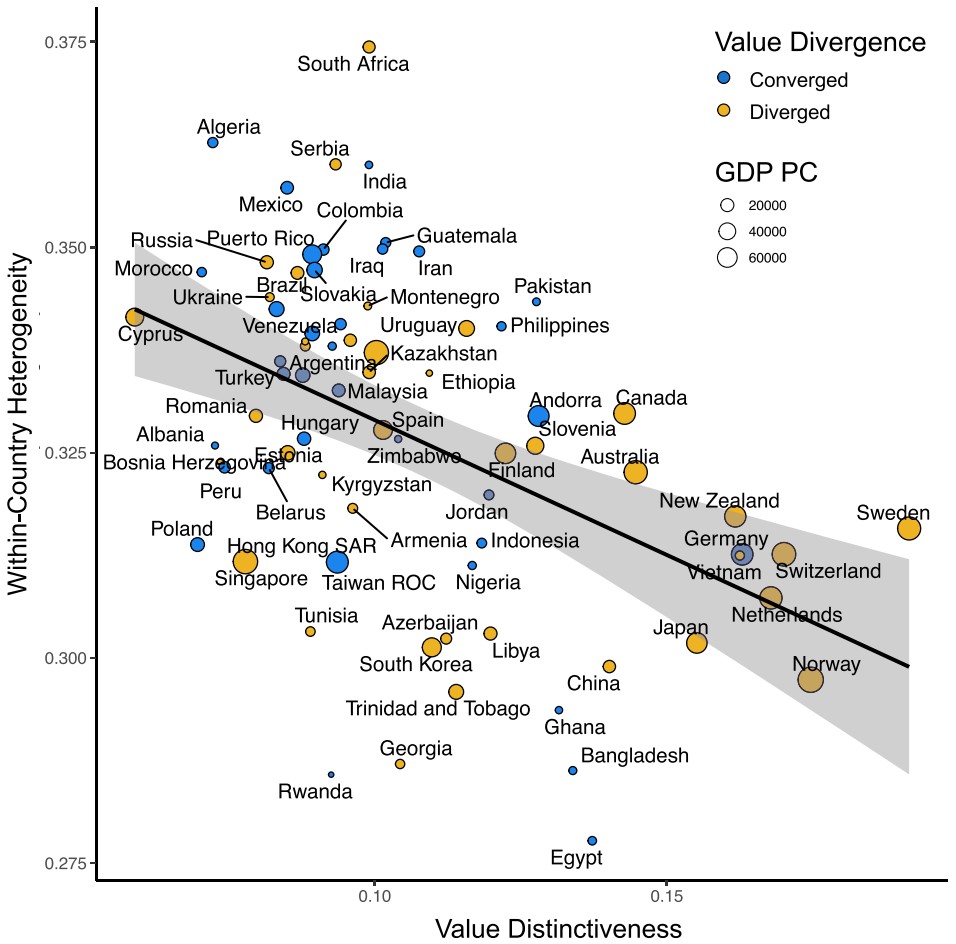

**Fig. 4 | Association between within-country heterogeneity and value distinctiveness.** The best fit line comes from a general linear model estimate, and the shading represents the standard error centered around the best fit line. Data-point size represents GDP per capita, and datapoint fill color represents whether country values have diverged (value distinctiveness has increased) or converged (value distinctiveness has decreased) over time.

GDP per capita is shown on the left of the y-axis. In this visualization, it is clear that most countries have diverged from the rest of the world, with more distinctive values at the final vs. the first timepoint. Second, in Western regions but not non-Western regions, divergence has been most stark for high-income countries.

These findings suggest a provocative possibility: rises in global wealth may be responsible for the worldwide divergence of values. Most countries have become wealthier over time—Western countries but also many non-Western countries like Singapore, Hong Kong, and South Korea[52]. This means that most countries were poorer in the earlier (vs. later) waves of the WVS. Western nations in these early waves held more emancipative values than non-Western nations, but not by a large degree. As time passed, rising wealth led Western countries to adopt more emancipative values, but it did not have the same effect for most non-Western countries. These trends led to a growing gap between high-income Western countries and the rest of the world.

Consider Hong Kong and Canada, where GDP per capita has followed a similar trend, but values have diverged. Both countries had a GDP per capita of approximately $25,000 in 2000, which doubled to approximately $50,000 by 2020[53,54]. Over the same time interval, beliefs that homosexuality was justifiable rose in Canada from 0.49/1.00 to 0.74/1.00. Perceived justifiability of homosexuality also rose in Hong Kong, but only from 0.29/1.00 to 0.44/1.00. This means that the gap in means between Hong Kong and Canada increased by 50% during this period. One of the fastest-changing values in Hong Kong

during this time was belief in children's work ethic. From 2000 to 2021, the percent of Hong Kong participants who mentioned responsibility as an important childhood quality rose from 19% to 52%, whereas it fell from 53% to 47% in Canada.

This example shows that wealth can sometimes lead to region-specific effects on values. If rising wealth fosters emancipative values in some regions but not in others, this could explain why Western democracies have developed more distinct values over time. This would be consistent with Eisenstadt, whose "multiple modernities" thesis states that countries follow their own trajectories of modernization[19]. It is also consistent with Huntington's observations that rising wealth and influence in East Asia could lead to a re-affirmation of traditional Confucian values[22]. Factors such as migration, political change, and decolonialization could also contribute to these trends. For example, sovereignty over Hong Kong transitioned from the United Kingdom to China in 1997, which may have affected people's values. More research is necessary to determine the causal relationship, if any, between wealth and value change.

### Within-country heterogeneity and value distinctiveness
In our next analysis, we explored whether value distinctiveness across countries was correlated with the heterogeneity of values within countries. In other words, are countries where citizens disagree on values more like the rest of the world than countries where citizens have more homogeneous values?

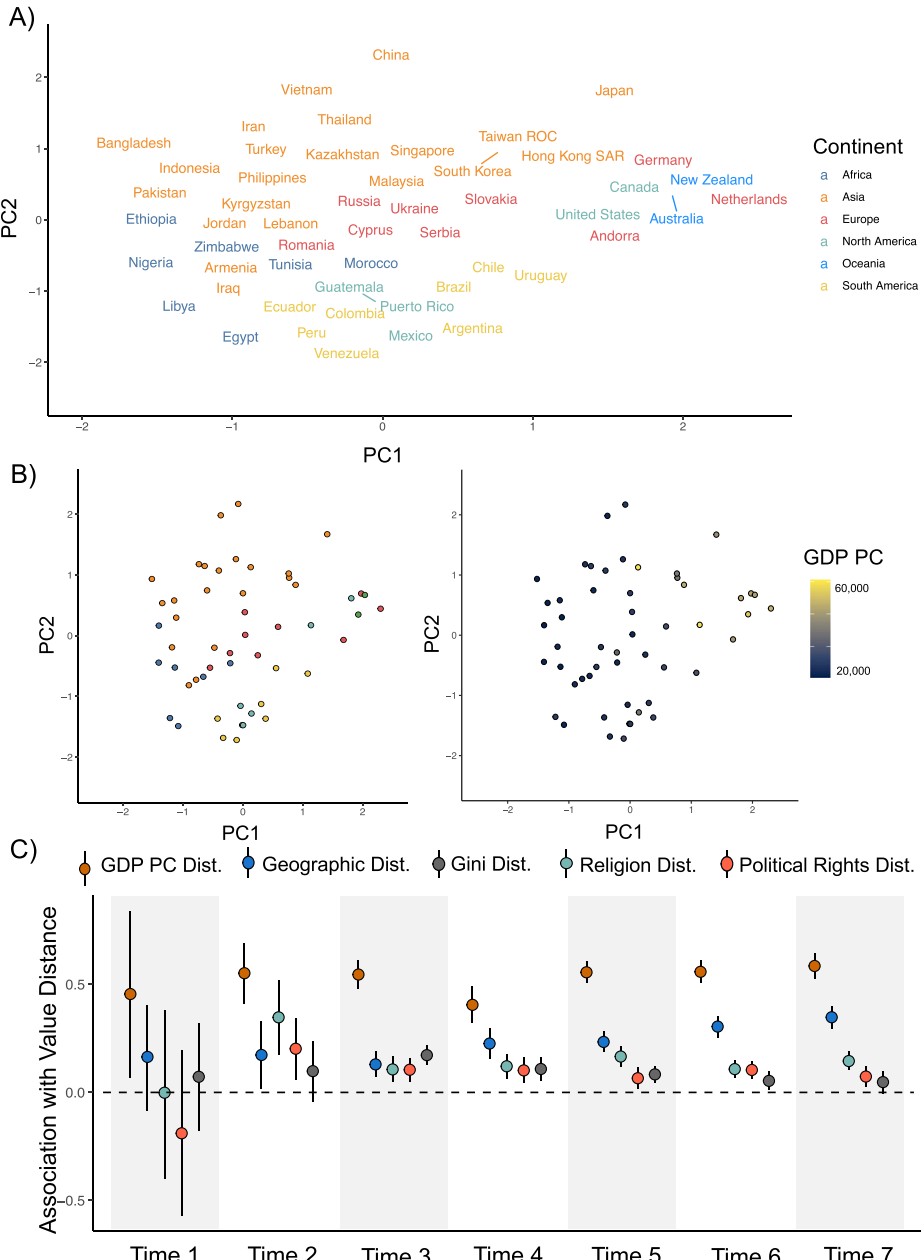

**Fig. 5 | Explaining value similarity across countries. A** Wave 7 country values projected on a two-dimensional space derived from a PCA of all 40 values. Proximity in this space indicates value similarity, and label color indicates continent. **B** A reproduction of (**A**) with countries as nodes colored by continent (left) and GDP per capita (right). **C** The association between pairwise differences in five geopolitical variables and value distance across nations, which involved $n = 11$ countries at Time 1, $n = 20$ countries at Time 2, $n = 48$ countries at Time 3, $n = 37$ countries at Time 4, $n = 52$ countries at Time 5, $n = 54$ countries at Time 6, and $n = 52$ countries at Time 7. Each node represents a regression coefficient, and these coefficients have been standardized so that they could be interpreted as effect sizes. Error bars are 95% confidence intervals centered around the regression coefficients.

We calculated within-country heterogeneity using the same approach that we used to calculate value distinctiveness. But instead of taking the difference of country means from the global mean across values, we took the deviation of individual ratings from the country mean. Higher levels of within-country heterogeneity therefore represent a country with high levels of disagreement among its citizens. For example, high within-country heterogeneity in the United States might arise because liberal and conservative Americans hold very different values. In exploratory analyses, we analyzed whether within-country heterogeneity was also rising across time, signaling a divergence within countries. However, we did not find general trends in this measure. Within-country heterogeneity seems to be rising among some countries and falling among others (see Supplementary Methods).

However, we did find that within-country heterogeneity was closely linked to value distinctiveness. When we regressed value distinctiveness on within-country heterogeneity controlling for GDP per capita, Gini, globalization, political rights, and distance from equator, within-country heterogeneity robustly and negatively predicted value distinctiveness, $b = -0.06$, $SE = 0.01$, $t(8,032) = -4.45$, $p < 0.001$, $\beta = -0.07$, 95% CIs [−0.09, −0.03]. In other words, countries where citizens agree more on values tend to be more different from other

countries. The other effects in this model remained unchanged. GDP per capita had a positive relationship and no other variables had significant associations (see Supplementary Table 12). Figure 4 visualizes the relationship between within-country heterogeneity and value distinctiveness. Our Supplementary Methods reports timepoint specific models.

These results suggest that as countries develop more homogenous values, they may increasingly splinter from viewpoints that are shared around the rest of the world. This has happened in high-income Western nations like Sweden and Norway, where there is a higher consensus around emancipative values such as justifiability of divorce and abortion. But it has also happened in poorer nations such as China and Ghana, where consensus has formed on different values. In contrast, countries with heterogeneous values such as South Africa and India may struggle with internal division, but when averaging across their subpopulations, these countries' values may resemble those of other countries.

**What country characteristics predict value similarity over time?**
In our final analysis, we explored which countries hold similar values to one another. We first fit a principal components analysis (PCA) to detect underlying dimensions in how the 40 values varied across nations. Two principal components (PCs) explained 80.70% of variation (see Materials and Methods). To understand the meaning of these dimensions, we regressed each dimension against previously developed dimensions of value variation: Welzel's secular values and emancipative values, and Inglehart's 12-item post-materialist index. PC1 was strongly positively linked to emancipative values, $b = 0.76$, $SE = 0.04$, $t(215.60) = 19.46$, $p < 0.001$, $\beta = 0.77$, $95\%$ $CIs$ [0.67, 0.84], and to secular values, $b = 0.26$, $SE = 0.03$, $t(233.90) = 8.60$, $p < 0.001$, $\beta = 0.26$, $95\%$ $CIs$ [0.20, 0.32]. PC2 had a smaller and less robust relationship with secular values, $b = -0.13$, $SE = 0.07$, $t(194.81) = -2.10$, $p = 0.042$, $\beta = -0.13$, $95\%$ $CIs$ [−0.27, 0.009]. No effects involving the choice index were robust (see Supplementary Methods for more information about these models).

We used these PCs as coordinates in a two-dimensional value space in which countries with similar values are closer together than nations with different values (see Fig. 5). We next fit regression models for each timepoint to test whether geopolitical characteristics could explain which countries were closer together in value space and which countries were further apart.

These regressions found that wealth has been the strongest correlate of value similarity across nations over time. High-income countries share values with other high-income countries, and poor countries share values with poor countries. This effect is visible in Fig. 5 (middle right), where high-income countries like New Zealand, Germany, the Netherlands, and Japan are clustered together on the right side of the value space. We also note that the left side of the value space in Fig. 5 is denser than the right side, which replicates our prior analysis: high-income countries have developed more dissimilar values compared to the rest of the world, primarily because of their unique endorsement of emancipative values.

We also found that geographic proximity has become progressively more correlated with value similarity over time, with an effect size rising from 0.16 (Timepoint 1) to 0.35 (Timepoint 7). This is visible in Fig. 5A: high-income East Asian countries like South Korea and Singapore are closer together than high-income Western nations like New Zealand and the Netherlands. The rising predictive power of geographic proximity suggests that values are converging within regions but are diverging across regions.

Religion has also emerged as a robust predictor of value similarity. Countries with more similar religious profiles have more similar values, even controlling for their similarity in wealth, geographic position, and other geopolitical features. This finding conceptually replicates a recent analysis that found that co-religionists—even those living in

similar countries—shared similar values[55]. The fact that values are segregating along geographical and religious fault-lines further supports Huntington's thesis that the 21st century would see a rise of ancestral cleavages in values. By the final WVS timepoint, religious similarity and geographic proximity were stronger correlates of value similarity than having similar levels of inequality or political rights. Figure 5 plots out the associations of value similarity with each form of geopolitical similarity at each timepoint. Table 3 lists key coefficients from the value similarity regressions.

## Discussion
We find that values have diverged from 1981 to 2022 across 76 countries. Value divergence appears to be strongest for values related to tolerance and openness and can be explained by the rising difference between high-income Western countries and the rest of the world. Regional value convergence has accompanied worldwide divergence, with geographically proximal countries adopting more similar values over time.

Our Supplementary Methods present many robustness tests that replicate our key findings with different sampling strategies, normalization procedures, and demographic weighting. They also include an additional literature review that summarizes previous research on modern cultural differences in values and other metrics of cultural distance in greater depth.

These robustness analyses lend more support to our conclusions, but there are still important limitations to this study. First, we can only claim that values have diverged across the 40 items in our analysis. These items include a wide range of different values. However, they are not an exhaustive list of values across countries, and our results could change if they were replicated on a different set of items. Second, our samples are not perfectly representative of populations in the countries that we have analyzed. The WVS attempts to emulate key dimensions of representativeness, but it may sometimes neglect responses from key demographics, such as indigenous populations. For these reasons, it will be important to replicate our analyses in new datasets. Few datasets have data on social values that match the WVS, but this may change as new cross-cultural surveys are launched and analyzed over time.

While acknowledging these limitations, we also view our findings as potentially impactful from both a theoretical and practical standpoint. Theoretically, our findings suggest that globalization and intergroup contact alone are not sufficient to produce converging social values. Our findings also suggest that the post-materialist hypothesis—that wealth breeds emancipative values and tolerance[8,9]—may have stronger predictive power in some regions than others. Our findings support key parts of other theories but do not completely align with any single theory of culture and values. We observe worldwide divergence of values accompanied by re-alignment of values along regional and religious fault-lines, which is consistent with Huntington's civilizations thesis and more recent work on rises in geopolitical regionalism from macroeconomics[56]. We also find that this effect of wealth varies across Western and non-Western countries, consistent with Eisenstadt's multiple modernities theory and with other studies that highlight Asia's unique modernization trajectory[57]. It may be that our findings are specific to a particular period of time following decolonization and the end of the Cold War (1981–2022) and that we would have found different results at different periods of time. Only time will tell if our findings represent a general cultural trend or a historically isolated phenomenon.

Value divergence could also explain theoretical puzzles in the social sciences. For example, there is a popular theory that rising wealth[4] and technology[58] facilitate religious decline because they decrease existential insecurity and relieve the economic pressure to have children[59]. But this model does not explain why rising wealth has not brought religious declines in Middle Eastern countries and has

**Table 3 | Effects and Confidence Intervals from Clustering Analysis Regressions**

| Timepoint | Association With Value Similarity ($\beta$ and 95% CIs) | | | | |
|---|---|---|---|---|---|
| | GDP PC | Geography | Religion | Gini | Political Rights |
| Time 1 | **0.46 [0.07, 0.84]** | 0.16 [−0.09, 0.40] | −0.001 [−0.40, 0.38] | 0.07 [−0.18, 0.32] | −0.29 [−0.57, 0.20] |
| Time 2 | **0.56 [0.41, 0.69]** | **0.17 [0.02, 0.33]** | **0.35 [0.17, 0.52]** | 0.10 [−0.04, 0.24] | **0.20 [0.06, 0.35]** |
| Time 3 | **0.55 [0.48, 0.61]** | **0.13 [0.07, 0.19]** | **0.11 [0.05, 0.17]** | **0.17 [0.13, 0.22]** | **0.11 [0.05, 0.16]** |
| Time 4 | **0.41 [0.32, 0.49]** | **0.23 [0.19, 0.28]** | **0.12 [0.06, 0.18]** | **0.11 [0.05, 0.16]** | **0.10 [0.04, 0.16]** |
| Time 5 | **0.56 [0.51, 0.61]** | **0.31 [0.26, 0.35]** | **0.17 [0.12, 0.21]** | **0.08 [0.04, 0.12]** | **0.07 [0.02, 0.12]** |
| Time 6 | **0.56 [0.51, 0.61]** | **0.31 [0.26, 0.35]** | **0.11 [0.07, 0.15]** | **0.05 [0.01, 0.10]** | **0.10 [0.06, 0.15]** |
| Time 7 | **0.59 [0.53, 0.64]** | **0.35 [0.30, 0.40]** | **0.15 [0.10, 0.19]** | 0.05 [−0.004, 0.10] | **0.07 [0.03, 0.12]** |

Statistically significant effects are denoted with bolded font.
All coefficients are standardized, which means that they can be compared with one another similar to *r* statistics. Full coefficients for these models are reported in Supplementary Table 17.

even correlated with rising religiosity in some of these countries. One possibility is that these declines will happen with time, especially with generational change. In other words, countries like Saudi Arabia and Qatar may not have been high-income long enough to experience secularization. However, another possibility is that Islam has emerged as a key part of Arabic post-colonial identity. Rising wealth and influence could have led Arab countries to emphasize the religious part of their identity to distinguish themselves from the West.

Our findings answer some important questions about contemporary value change, but they also pose new ones. Why has rising wealth led Western, but not non-Western, nations to espouse more emancipative values? We believe that this relationship might be embedded in ecological and cultural events throughout Western history, including the advent of participatory democracy in Athens and early Rome, enlightenment and post-enlightenment philosophy, the Catholic Church's marriage and family polices[60], the French revolution, and the reformation[61]. These phenomena may have gradually solidified a Western identity focused on autonomy, primacy of individual rights over obligations to the in-group, and tolerance for breaking norms[2,60,62]. As the world has globalized and Western nations have competed for resources on the world stage, this identity may have crystalized even further. But this does not mean that wealth or globalization should have encouraged similar values in African, East Asian, or South Asian cultures. To the contrary, growing power and resources may have prompted non-Western countries to affirm their own traditional values.

Practically, value divergence has implications for political polarization and conflict across world countries. Russia has framed the recent war in Ukraine as a war against Western values[40]. Chinese politicians have spoken against countries that "forcibly promote the concept and system of Western democracy and human rights"[63]. Western non-governmental organizations have faced recent accusations of seeding immorality and propagating Western imperialism[64], and public opinion polling has found increasingly hostile attitudes towards Western countries in the Middle East[65], Asia[66], and Africa[67]. Our findings do not shed light on the extent of the anti-Western sentiment or the exact nature of its antecedents and consequences. We do not know to what extent governments strategically propagate certain values to reinforce national identities, as Tomlinson[21] would suggest. One goal of future research could be to understand the extent that political elites have encouraged value divergence among ordinary citizens.

Our research also underscores the limitations of studying Westerners to make claims about human psychology writ large. Cross-cultural scholars have pointed out that people from Western, Educated, Industrialized, Rich, and Democratic (WEIRD) countries have psychological traits that differ from the rest of the world. This peculiarity presents an external validity problem for studies that recruit mostly WEIRD subjects, and it also presents an intellectual problem for cultural evolutionists who hope to explain regional variation. We show that this problem has become more acute in the last forty years. WEIRD subjects have become even more peculiar, at least in their social and moral values. This shift makes it more crucial than ever that behavioral scientists develop their theories using data from globally representative samples.

## Methods
Here we report key parts of our methodology and analysis plan. Our Supplementary Methods contain an extended materials and methods section with further information.

### Ethics
The director of the Social and Behavioral Science IRB office at University of Chicago determined that our research does not require IRB approval, since the nature of this research—analyzing archival deidentified data—does not quality as human subjects research.

### The World Values Survey
The World Values Survey (WVS) is an international research program devoted to measuring the social, political, economic, and religious values of individuals around the world using regular surveys. The WVS website contains comprehensive information about its research procedures (https://www.worldvaluessurvey.org). This includes information about translation procedures and fieldwork training. In addition to publishing data each wave, the WVS publishes a time-series file containing data from all waves. The WVS has not surveyed the same people over time in this file. Rather, each timepoint contains a demographically representative snapshot of people in a country at a particular point in time. The WVS also publishes a list of variables indicating which items are asked in different waves, and a list of countries indicating which countries are surveyed in each wave. The timeseries dataset is published in many different formats. We downloaded the Rdata format.

**Characteristics of countries.** We focused on the 76 countries where the WVS has collected data for at least two waves, a necessary condition since we are interested in change over time. Supplementary Table 1 summarizes the age-sex composition of each country in each wave, with dashes indicating waves where a country was not included in data collection. Samples are designed to be representative of people age 18 and older residing within private residences in each country, regardless of citizenship or language. The WVS employs probability sampling and stratified sampling to achieve these targets. They offer case weights to compensate for small deviations with respect to gender-age (self-reported), rural-urban, or educational attainment.

**Characteristics of Items.** The WVS contains a heterogeneous set of items that change each year. Of the 1,011 items included in the WVS

timeseries file, most were only included in a minority of waves. We focused on the items that were asked in all seven waves, and specifically on items that could be construed broadly as values. In total, we selected 40 items for analysis. Our extended Supplementary Methods summarize how we selected these items and excluded others. In Supplementary Table 2, we provide the item identification number, item label, and the scale that participants used to respond to the item. Readers can access the complete item wording for each item by downloading one of the PDF questionnaires for any wave from the WVS website.

One of the scales in our analysis involved the qualities that respondents felt to be important for children to learn. For these items, respondents were not allowed to select more than 5 items. However, during data processing we realized that this rule was not always followed. To keep the questionnaire format consistent across countries and waves, we excluded all respondents ($n = 20,380$; 5% of the total sample) who had selected more than 5 important childhood qualities. This decision did not affect our results. All results replicated with and without excluding participants who had not followed the instructions.

### Exogenous variables

We collected data on exogenous variables illustrating the geopolitical conditions of countries over time. We sought to match all exogenous variables as closely as possible to the year of WVS data collection, so we downloaded time-varying estimates of these variables. We provide the source and access URL for each variable here. Our extended materials and methods provide more information about each variable.

**GDP per capita.** We accessed GDP per Capita (current USD) from the World Bank. The access URL is https://data.worldbank.org/indicator/NY.GDP.MKTP.CD.

**Gini.** We accessed Gini coefficients from the World Inequality Database (WID). The access URL is https://wid.world/data/.

**Globalization.** We measured globalization using the widely used KOF index published by the Swiss Economic Institute. The access URL is https://www.theglobaleconomy.com/download-data.php.

**Political rights.** We measured political rights using the "Political Rights" index published by the Freedom House. We accessed this variable using the "Global Economy" database. The access URL is https://www.theglobaleconomy.com/download-data.php. We note that the Freedom House publishes two different indices: a political rights index and a civil liberties index. Both indices are coded on a 7-point scale from 1 (Strong) to 7 (Weak). These correlated at 0.94 and showed identical results, so we focused on the political rights index in our analyses. We recoded the index so that higher values meant more political rights.

**Religious distance.** We accessed data on the % of the population identifying as Hindu, Christian, Muslim, and Buddhist, and measured religious distance by summing the absolute difference in % of people identifying with each religious group across pairs of countries. The access URL is https://www.theglobaleconomy.com/download-data.php.

### Analytic strategy

This section summarizes how we computed value variation and value distinctiveness. Table 1 summarizes the meaning of these measures, along with within-country heterogeneity, which we describe in the main text.

**Item normalization.** Supplementary Table 2 shows that items were asked on different scales. Some items involved binary responses (e.g.,

whether people mention not wanting to be neighbors with someone from a specific group). Others were asked with Likert-type scales, such as the 1–10 scale that people used to rate whether behaviors were morally justifiable. We normalized the item scales using min-max normalization, which is a common approach in data science and machine learning[68]. Given a vector $\mathbf{V} = [v1, v2, ... vn]$, we can determine min_V as the minimum value in the vector and max_V as the maximum value in the vector. We can create our normalized vector, $\mathbf{V'}$ using:

$$v'_i = \frac{v_i - v_{\min}}{v_{\max} - v_{\min}} \qquad (1)$$

In other words, each element in the new vector is the result of subtracting the minimum value of the original vector from that element, then dividing by the range of the original vector (i.e., the difference between its maximum and minimum values). This procedure results in variables with ranges of 0–1, no matter their original scales. As an alternative to min-max normalization, we also considered a median split approach. We discuss the limitations of this approach in the Supplementary Methods, and show how our main findings replicate with this approach.

**Calculating value divergence across items.** After the item-normalization procedure, we took the mean values of items for each country. This resulted in country-level means for each item at each timepoint. As an exploratory analysis, we fit Pearson correlations between timepoint and mean for each item, which indicates how values are changing over time across all countries. Supplementary Table 3 lists the full set of trends in global mean endorsement of items over time.

We assessed value divergence across items by computing the standard deviation of country means at each WVS timepoint—the measure of "value variation" that we summarize in our introduction. We then estimated the linear trend in these SD values in a mixed effects model which we summarize in the main text, and also across a set of Pearson correlations that we fit for each individual item and that we summarize in Fig. 1. In these models, positive coefficients represent value divergence, since the SD of country means is increasing over time. In Supplementary Table 4, we report value divergence coefficients for four different subsets of countries— countries which were included in at least 2 WVS waves, 3 waves, 4 waves, and 5 waves. We conducted our analyses for these different subsets of countries because we wanted to ensure that value divergence was not due to changing WVS composition over time.

We calculated the relevance of each item to Welzel's emancipative and secular dimensions by taking the absolute value of the correlation between the item and each of the two dimensions to create continuous loading scores. Scores for both dimensions are published as variables in the WVS dataset. The item measuring participants' view of "unselfishness" as an important child quality had a loading of 0.01 on the emancipative values dimension and a loading of 0.003 on the secular values dimension, suggesting it had low relevance for both dimensions. In contrast, the item measuring participants' views on the justifiability of had a loading of 0.56 on the emancipative factor and a loading of 0.40 on the secular factor, reflecting higher relevance for both factors. The dimensions correlate at $r = 0.36$ with each other, suggesting that secular values tend to be more emancipative and that the two dimensions are not redundant.

**Calculating value distinctiveness.** We followed a multi-step process to calculate value distinctiveness for each country. First, we computed the global median score for each value at each timepoint. For example, the median score for the item "Important Child Qualities: Hard Work" was 0.55 in the third WVS wave. Next, we computed the absolute differences between each country's mean and the global median for each

item. For example, the average response to the "Hard Work" item in Albania was 0.57, yielding an absolute difference of 0.02. Finally, we aggregated across all of these absolute differences to obtain a "value distinctiveness" score for each country. This process of computing value distinctiveness $D_{i,j}$ can be expressed as:

$$D_i = \frac{\sum_{j=1}^{40} \left| n_{i,j - \text{median}\left(N_{1,j}, N_{2,j}, N_{3,j} \ldots N_{k,j}\right)} \right|}{40} \qquad (2)$$

Where $n_{ij}$ represents the mean value $j$ of a given country $i$. We used the median to compute the global value because it avoided outlier countries from having a large impact on the mean value.

Computing value distinctiveness across timepoints allowed us to estimate which countries have relatively unique values at any given time. In our main text, Fig. 3 summarizes each country's value distinctiveness score in the first and last wave that it was included in the WVS. Supplementary Fig. 2 visualizes value distinctiveness for each country in each WVS wave, and Supplementary Table 5 summarizes value distinctiveness for each country in each wave.

In addition to estimating the value distinctiveness of individual countries at specific timepoints, we also analyzed general trends in value distinctiveness over time. Our reasoning was that, if countries are diverging in their values, the average value distinctiveness coefficient would be increasing. This would indicate that countries are "spreading out" around the global medians of values. One limitation of this analysis is that the global midpoint is not truly "global"—it is only the average of the countries sampled by the WVS at a particular point in time. If the WVS has become systematically more diverse in its sampling, then this could artificially create value divergence via a trend in sample heterogeneity. This is why we repeated all of our analyses for subsets of countries that had participated in 2, 3, 4, and 5 WVS waves. It is also why we conducted the decade-over-decade analysis in which we replicated the finding when looking across a subset of 33 countries which provided data in the 1990s, 2000s, and 2010s—the three decades with the greatest WVS coverage. Supplementary Table 6 summarizes value distinctiveness scores for each decade and for each country within the 33-country sample.

**Clustering methodology.** In our final analysis, we sought to project countries onto an $n$-dimensional value space. Greater distance in this space between a pair of countries would represent a larger difference between the values of two countries. We could also test which geopolitical variables were most strongly correlated with this pairwise distance metric.

The first step in this process was to determine how many dimensions would appropriately capture sufficient variance across values. Supplementary Fig. 3 illustrates the correlations between all 40 items in our dataset. There were clear covariances across sets of values, and so we reasoned that a small number of dimensions might explain considerable variation across values. To determine the optimal number of dimensions, we fit a Principal Components Analysis (PCA) on the correlation matrix of values (the same matrix displayed in Supplementary Fig. 3). In this analysis, PC1 explained 65.4% of variation, and PC2 explained 15.3% of variation. No other dimension explained more than 10% of variation, so we adopted a 2-dimension solution. Supplementary Fig. 3 shows an elbow plot of variance explained by each PC, and the item loadings. We projected countries onto a two-dimensional value space using these PC item loadings multiplied by the country's scores on each value. This procedure generated the plots displayed in Fig. 5.

The regression that we describe in our main text (see Table 2) was fit to a dataframe where rows represented pairs of nations, with columns indicating each nation in the pairwise comparison. Subsequent columns indicated Euclidean distance in the two-dimensional value space, difference in GDP per capita, geographical distance between nations, etc. We fit a cross-classified mixed effects model with random effects for each nation in the pairwise comparison, and fixed effects entered simultaneously to control for covariation between geopolitical characteristics. We standardized all variables in the regression so that estimates were not influenced by the original measurement scales of the geopolitical variables. More positive values in these fixed effects suggest that countries with similar geopolitical characteristics also have more similar values. For example, a positive effect of GDP per capita would suggest that countries with very similar levels of wealth also have very similar social values. Changes over time in these coefficients would indicate that a geopolitical variable is becoming better or worse at predicting which countries had more similar values.

### Reporting summary
Further information on research design is available in the Nature Portfolio Reporting Summary linked to this article.

## Data availability
All data are available on the open science framework at https://doi.org/10.17605/OSF.IO/F9BZ7. Data on values can also be publicly downloaded from the World Values Survey at https://www.worldvaluessurvey.org/. Data on GDP per capita, annual GDP per capita growth, and the Gini coefficient were retrieved from the World Bank (https://data.worldbank.org/). Data on Gini coefficients was retrieved from the World Inequality Database (https://wid.world/data/). We retrieved data on globalization from the Swiss Economic Institute and data on political rights from the Freedom House. We used theGlobalEconomy.com, a dataset aggregator and supplier, to retrieve both datasets (https://www.theglobaleconomy.com/download-data.php). We also used theGlobalEconomy.com to retrieve our data on religious distance using the same link.

## Code availability
All code is available on the open science framework at https://doi.org/10.17605/OSF.IO/F9BZ7.

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

## Acknowledgements
We thank Rick Shweder, Eli Finkel, Michele Gelfand, Nour Kteily, and Nava Caluori for comments on an earlier draft of this paper. We thank the World Values Survey for providing high-quality, open-access time-varying data on values around the world.

## Author contributions
J.C.J. and D.M. both accessed the data, performed the analyses, and wrote the paper.

## Competing interests
The authors declare no competing interests.
