## [Peer Review File · Nature Communications]

Reviewers' Comments:

Reviewer #1:

Remarks to the Author:

This paper presents a fascinating set of results exploring whether countries are becoming more or less similar in their values over time. This is a question of great theoretical importance and one that is likely to be downright interesting to scientists in a variety of fields including psychology, anthropology, and political science. It will likely be of broad interest to lay audiences as well. The authors key findings are intriguing: 1) countries appear to be diverging rather than converging in their values over the past several decades, 2) this effect appears linked to wealth but more so or perhaps exclusively in Western countries, 3) Western societies are becoming more different from others over time, 4) spatial proximity seems to matter more for these trends now than in the past. Let me be clear I think this paper makes an important contribution and that the authors do a thorough and sophisticated job of analyzing the data.

That said, the presentation and somewhat inconsistent use of key terms sometimes make it difficult for the reader (or at least for this one) to assess how well the results substantiate key claims. I also think more nuance could be used in the introduction and that the authors might consider moving some portion of the results to the supplement as at present the reader may feel a bit overwhelmed. I believe that none of these are "fatal flaws," and that these issues can likely be addressed with additional analyses or clarification.

Below are some more detailed thoughts and suggestions. To be clear the length of my review shouldn't be taken as an indicator of my overall opinion of the paper. Rather these are points that I think could strengthen an already strong manuscript and make it clearer and easier to digest.

Intro

-Alienated seems a bit of a loaded term, perhaps replace with something more neutral?

-The authors discuss accounts of the origins of contemporary cultural differences as rooted in ecological conditions. They interpret these theories as suggesting that initial ecological conditions set the ball rolling so to speak for contemporary patterns of cultural variation which then became "canalized" over time. Although this seems to do justice to some of these accounts (i.e. Talhelm, et al., 2014; Thomson, et al., 2019), others (including some of studies cited in this portion of the intro like Grossmann & Varnum, 2015 or Varnum & Grossmann, 2016) in fact do not argue that it is not canalization or intergeneration transmission. Indeed at least two of the studies cited use time series data which imply that relatively contemporaneous ecological conditions may be responsible for cultural patterns. I'd suggest the authors nod to accounts that hold such impacts of ecology may not be confined to distal "initial conditions." Such accounts posit a role for evoked culture or adaptive phenotypic plasticity in explaining such links, and should be acknowledged here (i.e. Sng, et al., 2018; Rotella, et al., 2021).

-It's also not clear to me that the "forces that create cultural differences have waned over time." Perhaps biodiversity and modes of subsistence have become more homogenous around the globe, but I don't know if it's a completely fair characterization to say that all facets of ecology that effect culture have become essentially uniform around the world. Perhaps one could make this argument, but it would need to be better supported by evidence. I'd suggest toning down the strength of this claim here and acknowledging some of the nuance and uncertainty that surrounds this question . . .

Results and Methods

-The authors have two key measures, distinctiveness and divergence. They are defined sensibly in the intro, but are used somewhat interchangeably in the results. Given that these are conceptually

different measures, one tapping into how much variation there is within societies and the other how different scores for a country look from the global central tendency, this makes it a bit hard to interpret their results. This is especially the case when multi-level analyses are presented. I'd urge the authors to be a bit more consistent in their use of these terms and to more carefully unpackage the results so the reader can clearly follow. Similarly on page 11 the authors introduce another metric "value heterogeneity," which sounds a lot like "value variation," or what "divergence" as used in the first set of results presented. It feels like there is a lot of slippage in use of these terms that makes it hard to keep the story straight, at least for this reader.

-For example on page 8, the authors state "We next used centering to separate within-country and between-country trends. Worldwide value divergence was significant within countries, $b = 0.003$, $SE = 0.0005$, $t = 5.65$, $p < 0.001$, 95% CIs [0.002, 0.004], but not between countries, $b = 0.003$, $SE = 0.003$, $t = 0.93$, $p = 0.36$, 95% CIs [-0.004, 0.009]. This suggests that the same countries are growing farther apart over time, not that later vs. earlier WVS waves include more distinctive countries." Looking at these results and the table, my interpretation would be that there has not been increasing divergence in values between countries over time. Yet that's the opposite of the take home message of the paper. Maybe I'm missing something here, but if this results is consistent with the central finding of greater differences between societies in values over time, it needs to be more clearly explained how.

-The key finding here is that cultural values have become more divergent over time. However given the authors' operationalizations of value distinctiveness and variation and the fact that the number of countries including in early waves of the WVS was much smaller (and more heavily drawn from Europe and wealth societies) than in later waves, I wonder if part of this effect could be an artifact of the a greater number of countries from a greater number of regions being included in later waves of the WVS. In Wave 1 there were not 76 countries to my knowledge, but rather a much smaller set of 11 for example. There was also scant representation of countries in the global south in early phases of the WVS.

-In a similar vein, observing that Western countries showed more value divergence over time compared to other societies over time could reflect at least in part changes in the sample composition (in terms of which societies were included) of the WVS over time . . .

-The authors do present several analyses designed to assess these concerns in the supplement (i.e. Table S4) and pages 37-40., but they should better unpackage how this rules out an artifactual explanation

-On pages 12-13 the authors present evidence that spatial proximity explains more similarity in values now than in the past. This is an intriguing finding, but here too I wonder whether the fact that the sample of countries in early vs. later waves differs so greatly might be playing a role. At minimum it would be helpful to see if spatial autocorrelation among countries included in earlier waves was comparable to those in later waves. As far as I can tell the key analyses don't control for spatial autocorrelation, which would be important to do potentially.

-Michael E. W. Varnum

Reviewer #2:

Remarks to the Author:

This paper presented a series of analyses of World Values Survey data, which indicate that worldwide endorsement of cultural values have become more variable and divergent over the last 40 years. This study addresses important big questions about global changes in culture, by finding a novel way to approach the wealth of information contained within the World Values Survey. While this is not the

only method that could be used to address the underlying theoretical claims, it is certainly an appropriate and interesting approach that can compliment other methods (such as those that rely on other dimensions of cultural traits or more specific case studies). The data and code that they present could also be valuable for researchers who want to test alternative explanations for how countries are diverging over time, to further build of the current findings.

The present analyses provide a nice snapshot of overall trends across a wide variety of cultural values and a comprehensive set of countries, which suggests a broad pattern that future researchers could investigate using more targeted investigations of specific issues or locations. The authors also do a good job of stress-testing the findings in several ways, such as using multiple indicators of cultural divergence, analyzing different sub-sets of countries, and testing the variable effects across different items, and find compellingly consistent findings across different ways of examining the data. Overall, I believe this paper would be of interest to many researchers across several disciplines in the social sciences.

The visualizations are excellent – they are quite complicated, which means they can take longer to interpret, but they also contain a lot of rich information by layering several variables into the figures, and therefore provides even more insight into the results.

The two measures that they use to indicate cultural divergence – “value variation” and “value distinctiveness” – are both plausible ways to index the strength of divergences across different countries. The authors do a good job explaining conceptually how these measures are calculated in the introduction, so that readers can understand the results, while providing more details in the methods and supplemental materials which would allow interested researchers to replicate their analyses. Because these metrics are not typical in pre-existing literature, it was a bit hard at first to parse how these metrics were distinct from one another, and in the results I was sometimes confused about which metric was being referred to in each analysis or figure. For example, in Figure 1 the value is simply labeled “Divergence” on the x-axis, but I think this is derived from value variation, which is something that could be explained in the figure caption or on the figure itself. I also found it sometimes unclear what the predictor or outcome variables were in the regression analyses described briefly in the results. Adding sub-headings in the results section (if allowed by journal style guides) could also help distinguish when analyses shift from focusing on one indicator of cultural divergence to the other. The authors already do a decent job at explaining their measures and analyses, but a little more hand holding could help readers, especially in the results section. Their breaks in the results to summarize the overall pattern and what it might mean are helpful to keep the reader on track, but being even more specific in the figure caption or regression descriptions could be helpful.

In regression equations predicting country-level value distinctiveness from a variety of plausible variables (e.g., inequality, globalization, political rights), GDP was consistently a strong predictor, and often only the significant one. However, I wonder how strongly GDP is correlated with other predictors in the model, and therefore might be artificially sucking up variance from other predictors (e.g., rich countries might also be more globally connected and have more political rights).

The authors cluster the data into a 2-dimensional structure using PCA, which they then use to calculate the distance between countries across many different variables. Previous authors have also tried to provide a summary of the underlying structure across many WVS items (e.g., the Inglehart-Welzel cultural map of Traditional values vs. Secular-rational values and Survival values vs. self-expression values). Are the presently-identified dimensions comparable to the underlying value dimensions identified by previous researchers? Discussing this could help draw clearer connections between the present paper and prior research.

Reviewer #3:

Remarks to the Author:

This is an excellent paper on a very interesting topic and an important contribution to the behavioral and social sciences. The authors of the paper are asking a simple and important question – is the world converging or diverging in cultural values? The question is simple but answering it conclusively is no easy task. The authors have done an excellent job of drawing on multiple waves of data from the World Values Survey, using the most advanced statistical methods, and conducting several robustness checks. The paper is written for a broad audience.

The authors have done a good job of ruling out the biggest counter explanation, that these trends are artifacts of the WVS sampling getting more diverse over time. They rule this out in several ways. This is a big strength of the study.

The authors do a very good job providing context on the different theories that offer competing predictions regarding the divergence or convergence of worldwide values. Below are suggestions, questions, and recommendations, with the goal of strengthening the paper's contribution and potential impact.

- Given the role “emancipative values” have in the results, it might be worth it to break down emancipative values and Welzel's theory in greater detail. It was not entirely clear what an emancipative value was. For example, some research showing there has been a “global rise of individualism” is discussed alongside a study that showed “emancipative values are diffusing around the world” but the distinction between the two is not clear to me. I think this problem is present throughout the manuscript, given how much emancipative values are discussed, but a sharper conceptual analysis and discussion could deal with this problem.
- It would be helpful to clarify how the authors are conceptualizing the relationship between the “value distinctiveness” and “value variation” measures. Are these two measures assessing the exact same construct but in different ways? In this sense, is the value distinctiveness measure a robustness check on the results using the “value variation” measure, or are these two measures measuring subtly different aspects of the construct? My impression was that these are two different conceptualizations of the same construct, and that “value distinctiveness” is essentially a robustness check for the “value variation” measure, but more clarity would be useful and important.
- Complementing on these two measures of calculating divergence, there is a third method that could be used to significantly bolster the paper's robustness checks. Get cultural distances by calculating average CFst of all countries per wave, then find out if cultural distances are growing over time, which would indicate divergence. See Muthukrishna et al., 2020 for this method.

M. Muthukrishna et al., Beyond Western, Educated, Industrial, Rich, and Democratic (WEIRD) psychology: Measuring and mapping scales of cultural and psychological distance. *Psychological Science*, 31, 678–701 (2020).

- The results section was relatively clear, and the figures and tables the authors chose to include in the main text are well done. The discussion is also very well written and contextualizes many of the findings.
- The GDP per capita analyses require a little more GDP context. For example, what is the statistical power for the analyses showing that “the effect of GDP per capita was significantly weaker in Africa and Asia compared to the effect in Europe” or that “wealth was not associated with value distinctiveness in African and Asian countries”. The WVS is a massive dataset, and these analyses utilized MLM's, so it's entirely possible they are well-powered, but given that it appears these comparisons are occurring at the country-level, some discussion of statistical power is necessary. This would help readers determine how much stock they should put into these subset analyses, given that there doesn't appear to be a good explanation for why this is the case.
- The comparison of Australians' changing opinions on childhood obedience compared to the changing opinions of Indians's was very helpful in providing some texture to some of the effects. Is the growing divergence driven by some countries remaining constant in their values over time, while Western

“progressive” countries are undergoing change? This example suggests that it is countries going in opposite directions. A few more examples of this kind for some of the highest changing values would be helpful. A table of some sort providing a few more example items and changes in responses from Wave 1 to Wave 7 for prototypical countries would go a long way to contextualizing these trends.

- p. 8 “This pattern of results is consistent with the illustration in Figure 2 that values have diverged non-linearly. These analyses address the concern that value divergence is an artifact of changing country membership in the WVS over time; value divergence is visible even when the sample of countries is held constant over time.”

What is the tipping point? It was not explained and it’s not clear from the figure (there seem to be multiple inflection points?)

- p. 11 “We found that people within countries are not diverging in values as much as countries are diverging from each other”

Does this mean nation-states are becoming more significant cultural forces over time, not less? (That is quite the opposite of the idea that the nation-state concept is being torn apart by tribalism on one end, and globalization on the other).

- p. 14. “Religion has also emerged as a robust predictor of value similarity. Countries with more similar religious profiles also have more similar values, even controlling for their similarity in wealth, geographic position, and other geopolitical features.”

This is good to know and it is a replication of White, Muthukrishna, & Norenzayan, 2021, PNAS, that is not cited or mentioned. This effect of religion on cultural values remains holding constant wealth and country.

- Also p. 14. “Value divergence appears to be strongest for values related to tolerance and openness and may be explained by rising differences between wealthy Western countries and the rest of the world. Worldwide value divergence has been accompanied by regional convergence, with geographically proximal countries adopting more similar values over time.”

The findings in this paper suggest that Western countries, already extreme outliers to begin with, are even more so now than they were a few decades ago. Does this not suggest that the WEIRD problem has become more acute over time? This has deep implications for those behavioral and social science fields that overwhelmingly skew WEIRD in their samples. Are social scientists studying cultural bubbles that are becoming even more isolated from worldwide patterns?

- In the paragraph on page 15 discussing the practical implications of value divergence for “political polarization and conflict across world countries”, it was unclear why right-wing populism is a trend consistent with the findings. Aren’t most of the right wing populism movements occurring in wealthy Western nations over the past ~10 years? If anything, wouldn’t this suggest that wealthy Western nations may be backsliding into more traditional values consistent with the rest of the world?

Moreover, political polarization consists of within-country divergences, of which there was limited discussion in this manuscript (aside from a few aspects of the results section). The authors should either make the link between worldwide divergence in values and within country divergence more clear, or just remove some aspects of this paragraph. I think part of the confusion here stems from this paragraph seemingly contradicting the findings that within-country homogeneity was associated with greater country divergence. In this sense, the widening divergence in values on the worldwide level is somewhat in conflict within the seemingly widening divergence within countries as well.

- Figure 4 shows that the more homogenous a country is (regardless of emancipative or traditional values), the more distinctive they are relative to the world. But this doesn’t seem to fit with the pattern we see that Western democracies are becoming more culturally diverse over time. Even countries at the extreme end (Norway, Netherlands) are more ethnically and culturally diverse than they used to be a few decades ago.

- p. 15. “Why is wealth associated with value distinctiveness in Western but not non-Western countries? One possibility is that historical events like the Protestant Reformation⁴⁷ or the Catholic Church’s Marriage and Family polices⁴⁸ predisposed the West to endorse emancipative values to a greater extent than other world regions.”

Again, are the authors treating “emancipative values” as the same thing as individualism (which is the theory cited here)?

- What about the late sociologist Ronald Inglehart’s recent findings in his latest book (Religion’s

Sudden Decline), that looking at all the waves of the WVS, we see that except for the Muslim world, and possibly some segments of post-communist countries (e.g., Russia), greater existential security (that is wealth combined with widely available social safety nets) has led to a decline in religiosity. This is also happening in the United States, which for decades was resisting the worldwide trend. How are the findings reported here related to these secularization findings?

Reviewer #1 (Remarks to the Author):

- 1. This paper presents a fascinating set of results exploring whether countries are becoming more or less similar in their values over time. This is a question of great theoretical importance and one that is likely to be downright interesting to scientists in a variety of fields including psychology, anthropology, and political science. It will likely be of broad interest to lay audiences as well. The authors key findings are intriguing: 1) countries appear to be diverging rather than converging in their values over the past several decades, 2) this effect appears linked to wealth but more so or perhaps exclusively in Western countries, 3) Western societies are becoming more different from others over time, 4) spatial proximity seems to matter more for these trends now than in the past. Let me be clear I think this paper makes an important contribution and that the authors do a thorough and sophisticated job of analyzing the data.**

That said, the presentation and somewhat inconsistent use of key terms sometimes make it difficult for the reader (or at least for this one) to assess how well the results substantiate key claims. I also think more nuance could be used in the introduction and that the authors might consider moving some portion of the results to the supplement as at present the reader may feel a bit overwhelmed. I believe that none of these are “fatal flaws,” and that these issues can likely be addressed with additional analyses or clarification.

Below are some more detailed thoughts and suggestions. To be clear the length of my review shouldn't be taken as an indicator of my overall opinion of the paper. Rather these are points that I think could strengthen an already strong manuscript and make it clearer and easier to digest.

Thank you for this extremely helpful review! We are glad that you found this manuscript interesting and important, and each of your comments led us to make key changes in the paper in order to make the findings and the interpretation stronger. Among these changes:

- We have revised our introduction to be clearer about the role of ecology in producing, sustaining, and changing cultural differences.
- We have renamed our measures so that they can be more easily differentiated by readers.
- We have clarified and foregrounded the method in which we control for changing WVS sample composition over time, and have added additional analyses that control for spatial autocorrelation when we compute value divergence.

We describe these changes in greater detail below. We hope that you enjoy the revised manuscript. Thank you again for the helpful feedback!

Intro

2. Alienated seems a bit of a loaded term, perhaps replace with something more neutral?

Thank you for pointing this out. We agree, and we have made the language more neutral in our revision. For example, the phrasing in our abstract is now “wealthy Western countries have developed values that are especially dissimilar from the rest of the world.” We have also removed the terms “alienated” and “alienation” from other sections based on your feedback.

3. The authors discuss accounts of the origins of contemporary cultural differences as rooted in ecological conditions. They interpret these theories as suggesting that initial ecological conditions set the ball rolling so to speak for contemporary patterns of cultural variation which then became “canalized” over time. Although this seems to do justice to some of these accounts (i.e. Talhelm, et al., 2014; Thomson, et al., 2019), others (including some of studies cited in this portion of the intro like Grossmann & Varnum, 2015 or Varnum & Grossmann, 2016) in fact do not argue that it is not canalization or intergeneration transmission. Indeed at least two of the studies cited use time series data which imply that relatively contemporaneous ecological conditions may be responsible for cultural patterns. I’d suggest the authors nod to accounts that hold such impacts of ecology may not be confined to distal “initial conditions.” Such accounts posit a role for evoked culture or adaptive phenotypic plasticity in explaining such links, and should be acknowledged here (i.e. Sng, et al., 2018; Rotella, et al., 2021).

You are right about this. We do not believe that exclusively intergeneration transmission and distal socioecological factors can drive cultural variation.

We have now updated our discussion of ecology and culture to explicitly note the possibility of rapid culture-ecology co-evolution, and we also nod to evoked culture accounts (and added a cite to the Sng paper, which we both admire). We have pasted the most relevant paragraph below:

Evolutionary models also offer mixed predictions for value convergence. According to many of these evolutionary models, cultural differences can arise from socioecological pressures involving subsistence style²³, population pressure⁷, resource scarcity², climate²⁴, and pathogen load^{25,26}. Cultural values and norms often promote behavior that is adaptive in light of these pressures²⁷. Sometimes these values and norms emerge over long periods of

history, but they also can change quickly^{25,26}, which evolutionary psychologists have termed “evoked culture”²⁸.

We hope that this conveys a less one-sided view of how ecology influences culture.

4. **It’s also not clear to me that the “forces that create cultural differences have waned over time.” Perhaps biodiversity and modes of subsistence have become more homogenous around the globe, but I don’t know if it’s a completely fair characterization to say that all facets of ecology that effect culture have become essentially uniform around the world. Perhaps one could make this argument, but it would need to be better supported by evidence. I’d suggest toning down the strength of this claim here and acknowledging some of the nuance and uncertainty that surrounds this question . . .**

Thanks for pointing this out. The question of whether ecology has become more uniform is not clear to us either, and we apologize for coming across more strongly than we intended in our original draft.

We rewrote the paragraph and now explicitly say that, while some factors might suggest that ecologies have become more similar, there has been no conclusive study that addressed all the facets of ecology (and it is not even clear whether such a study would be possible given the many ways in which ecology can vary). In general, we try to be much more even-handed in that paragraph:

Given the co-evolution between culture and ecology, one might expect that values should converge if people’s environments have also become more similar. In some specific ways, environments do seem to be more similar. The decline of biodiversity and the diffusion of new technology mean that people around the world consume the same foods and use the same products with greater ease than ever before^{21,29}. But it is not actually clear whether environments have become more homogenous around the world, because socioecological diversity is so multidimensional and difficult to quantify. Socioecological convergence also seems to be more pronounced within geographic regions rather than across regions. Studies have found that countries in the same trading blocs have developed more similar economic, demographic, financial, and political characteristics over time, whereas countries from different trading blocs have not become more similar⁴¹.

Results and Methods

5. **The authors have two key measures, distinctiveness and divergence. They are defined sensibly in the intro, but are used somewhat interchangeably in the results. Given that these are conceptually different measures, one tapping into how much variation there is within societies and the other how different scores for a country look from the global central tendency, this makes it a bit hard to interpret their results. This is especially the case**

when multi-level analyses are presented. I'd urge the authors to be a bit more consistent in their use of these terms and to more carefully unpackage the results so the reader can clearly follow. Similarly on page 11 the authors introduce another metric "value heterogeneity," which sounds a lot like "value variation," or what "divergence" as used in the first set of results presented. It feels like there is a lot of slippage in use of these terms that makes it hard to keep the story straight, at least for this reader.

We agree that these terms are conceptually similar and there is potential for confusion which we should address more earnestly. Your feedback echoes a point by Reviewer 3, who noted that *"it would be helpful to clarify how the authors are conceptualizing the relationship between the "value distinctiveness" and "value variation" measures. Are these two measures assessing the exact same construct but in different ways?"*

We have made several revisions to try to make these measures as clear as possible.

12. We have moved up Table 3 ("Description of Key Measures") from our original submission to the introduction, where it is now Table 1. This table juxtaposes the different measures in our paper and provides their definitions. We hope that it serves as an easy point of reference for readers.
13. We have renamed "value heterogeneity" to be "within-country value heterogeneity". This updated name communicates that the measure is assessing how values vary across people in a country, rather than across countries.
14. We have explicitly noted in the introduction that value variation and value distinctiveness represent the same outcome—convergence vs. divergence of values across world nations. Value variation allows us to test for divergence at the level of the item (e.g., in Figure 1), whereas value distinctiveness allows us to test for divergence at the level of the country (e.g., in Figure 3).
15. We have incorporated subheadings into the results section which clearly distinguish between analyses using testing value divergence at the item level (value variation) and at the country level (value distinctiveness)

Here is the paragraph introducing the measures in our revised manuscript:

In this work, we develop a general method to test whether social values have diverged or converged across the 76 countries that have provided multiple waves of WVS data. Instead of focusing on the mean levels of particular values over time, as in most prior studies, we create metrics that explicitly measure variation in values (see Table 1). These measures are meant to represent the same outcome—convergence vs. divergence of values across world nations. However, they vary in their level of analysis. The first measure, "value variation," focuses on divergence at the level of the WVS item whereas the second measure, "value distinctiveness," focuses on divergence at the country level. Table 1 describes both measures and defines a third measure called

“within-country heterogeneity,” which we use in secondary analyses to represent the homogeneity vs. heterogeneity of values within a nation. We define all three measures here so that readers can distinguish between them in our presentation of results.

And here is the table that we have moved to the introduction to clearly define the measures:

Table 1. Descriptions of Key Measures			
Measure	Unit of Analysis	Methodology	Interpretation
Value Variation	Item	We normalize responses to the 40 social value items that have been included in each WVS timepoint to a 0–1 scale. We then compute the standard deviation for the global distribution of country means at each timepoint. Higher standard deviations represent more value variation. We can calculate this trend for a single item, or across all items.	A rise in item-level value variation over time indicates value divergence, whereas a decline indicates convergence.
Value Distinctiveness	Country	After normalizing item responses and calculating country means for each item, we take the median value of these country means. This represents the global average of a given item at a particular point in time. We then take the absolute difference between each country mean and the global average. Higher values of this absolute difference mean that a countries’ values are dissimilar than most other countries.	Countries with high value distinctiveness across all values are dissimilar from countries in the rest of the world. If value distinctiveness increases in a country over time, this suggests that values in that country are diverging from those of other countries. If value distinctiveness is rising across all countries, this would indicate that countries are becoming more dissimilar from each other over time, indicating global value divergence.
Within-Country Heterogeneity	Country	We calculated within-country heterogeneity by applying the same procedure for estimating between-nation value variation to estimate variation of values across	Within-country heterogeneity measures whether individuals within countries hold different values. For example, high within-country heterogeneity in the United States might arise

		people living in the same nation.	because liberal and conservative Americans hold very different values.
--	--	--	---

Thank you for encouraging us to communicate our results more clearly.

6. For example on page 8, the authors state “We next used centering to separate within-country and between-country trends. Worldwide value divergence was significant within countries, $b = 0.003$, $SE = 0.0005$, $t = 5.65$, $p < 0.001$, 95% CIs [0.002, 0.004], but not between countries, $b = 0.003$, $SE = 0.003$, $t = 0.93$, $p = 0.36$, 95% CIs [-0.004, 0.009]. This suggests that the same countries are growing farther apart over time, not that later vs. earlier WVS waves include more distinctive countries.” Looking at these results and the table, my interpretation would be that there has not been increasing divergence in values between countries over time. Yet that’s the opposite of the take home message of the paper. Maybe I’m missing something here, but if this result is consistent with the central finding of greater differences between societies in values over time, it needs to be more clearly explained how.

We apologize for the unclear language. This confusion stems from two different meanings of “within-country”.

In a mixed effects model, it is possible to use group centering to separate cohort effects from longitudinal effects (Enders & Tofighi, 2007, *Psychological Methods*). A cohort effect would mean that value divergence is an artifact of later waves of the WVS containing more value-distinct countries than earlier waves (e.g., Vietnam is a distinct country, and it is only included in waves 4-6), whereas a longitudinal effect would mean that countries are becoming more distinct over time (e.g., Vietnam is becoming more distinct from other countries over time). This centering approach is a common approach using mixed effects models, and it is typical to call the longitudinal effect a “within-country” effect and the cohort effect a “between-country” effect.

In the context of our paper, however, this terminology is confusing because “within-country” value divergence could also refer to divergence of the values of individuals living within a country, like if liberal and conservative Americans were diverging in their values over time. This second meaning represents the construct we now label as “within-country heterogeneity.”

To avoid any confusion, we have changed our terminology. We now use the terms “cohort effects” and “longitudinal effects.” Here is the revised language:

One concern is that these findings are confounded with cohort effects: If the cohorts of the WVS are becoming increasingly diverse, then the mean level of value distinctiveness would rise even if countries’ values stayed consistent over time. We took several steps

to address this concern. First, we used centering to separate each country's general value distinctiveness across all waves (representing a cohort effect) from its change over time from wave to wave (representing a longitudinal effect). A model including both variables found that the longitudinal effect was significant, $b = 0.003$, $SE = 0.0005$, $t(10,520) = 5.65$, $p < 0.001$, 95% CIs [0.002, 0.004], but the cohort effect was not significant, $b = 0.003$, $SE = 0.003$, $t(45.19) = 0.93$, $p = 0.36$, 95% CIs [-0.004, 0.009]. This suggests that the same countries were diverging in their values over time.

We have also moved the separation of between and within effects in Table 1 to the supplemental materials (see the section "Separating Longitudinal and Cohort Effects with Centering" and Supplementary Table 9). We think that these results are useful for interpreting the findings, but they may introduce terminology confusion in a paper that is already quite complex.

Thank you for the feedback. We hope that it is now clearer why these findings support the main result of our paper.

Enders, C. K., & Tofighi, D. (2007). Centering predictor variables in cross-sectional multilevel models: a new look at an old issue. *Psychological Methods*, 12(2), 121-138.

7. **The key finding here is that cultural values have become more divergent over time. However given the authors' operationalizations of value distinctiveness and variation and the fact that the number of countries including in early waves of the WVS was much smaller (and more heavily drawn from Europe and wealth societies) than in later waves, I wonder if part of this effect could be an artifact of the a greater number of countries from a greater number of regions being included in later waves of the WVS. In Wave 1 there were not 76 countries to my knowledge, but rather a much smaller set of 11 for example. There was also scant representation of countries in the global south in early phases of the WVS.**

In a similar vein, observing that Western countries showed more value divergence over time compared to other societies over time could reflect at least in part changes in the sample composition (in terms of which societies were included) of the WVS over time . . .

The authors do present several analyses designed to assess these concerns in the supplement (i.e. Table S4) and pages 37-40., but they should better unpackage how this rules out an artifactual explanation

On pages 12-13 the authors present evidence that spatial proximity explains more similarity in values now than in the past. This is an intriguing finding, but here too I wonder whether the fact that the sample of countries in early vs. later waves differs so greatly might be playing a role. At minimum it would be helpful to see if spatial autocorrelation among

countries included in earlier waves was comparable to those in later waves. As far as I can tell the key analyses don't control for spatial autocorrelation, which would be important to do potentially.

Thank you for this important comment. You are absolutely right that accounting for the changing sample composition of the WVS is crucial for our analysis. The WVS has changed in its sample over time, recruiting an increasing number of developing countries. As you point out, Wave 1 was relatively small compared to the other waves.

In our original submission and revision, we conducted multiple analyses which were designed to rule out value divergence as an artifact of changing sample membership. We try to feature these analyses prominently in our revision, based on your feedback.

We begin with the centering analysis, described in response #5, that distinguishes cohort effects from longitudinal effects:

One concern is that these findings are confounded with cohort effects: If the cohorts of the WVS are becoming increasingly diverse, then the mean level of value distinctiveness would rise even if countries' values stayed consistent over time. We took several steps to address this concern. First, we used centering to separate each country's general value distinctiveness across all waves (representing a cohort effect) from its change over time from wave to wave (representing a longitudinal effect). A model including both variables found that the longitudinal effect was significant, $b = 0.003$, $SE = 0.0005$, $t(10,520) = 5.65$, $p < 0.001$, 95% CIs [0.002, 0.004], but the cohort effect was not significant, $b = 0.003$, $SE = 0.003$, $t(45.19) = 0.93$, $p = 0.36$, 95% CIs [-0.004, 0.009]. This suggests that the same countries were diverging in their values over time.

We next restrict our sample to countries that have participated in more waves of the WVS. This reduces interference associated with changing WVS composition over time, because it excludes countries which only appeared in a small number of WVS waves. For example, when we focus on countries which have participated in at least 5 WVS waves, we analyze a smaller sample of 18 countries which is highly stable over time, making it unlikely that our results stem from more diverse countries appearing in later WVS waves.

We also conducted more conservative tests in which we recomputed value variation and replicated our models among subsamples of countries that had less turnover and hence less susceptibility to cohort changes. We replicated value divergence among the 54 countries that participated in at least 3 waves, $b = 0.003$, $SE = 0.0006$, $t(8,596) = 5.28$, $p < 0.001$, 95% CIs [0.002, 0.004], the 32 countries that participated in at least 4 waves, $b = 0.003$, $SE = 0.0006$, $t(6140) = 4.59$, $p < 0.001$, 95% CIs [0.001, 0.004], and the 18 countries that participated in at least 5 waves, $b = 0.003$, $SE = 0.0007$, $t(4,060) = 4.42$, $p < 0.001$, 95% CIs [0.002, 0.005].

Finally, we go one step further by creating a sample of 32 countries which provided data in the 90s, 00s, and 10s. We replicate our findings with this 32-country sample, which

shows that ***we can detect value divergence even when there are no changes to the sample of countries over time.*** In our view, this is strong evidence that value divergence is not an artifact of changing WVS sample composition over time.

We then replicated the finding in sample with no turnover in sample composition—a subset of 32 countries that provided data in the 1990s, 2000s, and 2010s, which were the three decades with the greatest WVS coverage. Value distinctiveness was higher in the 2000s, $b = 0.01$, $SE = 0.003$, $t(78) = 4.17$, $p < 0.001$, 95% CIs [0.007, 0.02], and the 2010s, $b = 0.01$, $SE = 0.003$, $t(78) = 4.10$, $p < 0.001$, 95% CIs [0.007, 0.02], compared to the 1990s—although there was no significant difference between the 2000s vs. 2010s, $b = -0.0002$, $SE = 0.003$, $t(39) = -0.08$, $p = 0.94$, 95% CIs [-0.006, 0.005]. This decade-based analysis provides evidence for value divergence even when we fix the country sample over time.

These analyses are all described on page 9 of the main text, and then revisited in greater depth within our supplemental materials.

To supplement these analyses, we now also control for spatial autocorrelation over time in our analyses of value distinctiveness. In our original submission, we controlled for autocorrelation using continent random effects in our multi-level models. Specifying continent random effects is an approximate way of controlling for spatial autocorrelation because countries in the same continent are treated as interdependent datapoints, but this method is imperfect because it does not adjust for variation in the proximity of countries within continents. For example, India, Japan, and China are all Asian countries, but China and Japan are more similar than either country is to India.

We now compute continuous spatial autocorrelation using the same approach that we use for computing value distinctiveness. We calculate the “average” geographic coordinates of each WVS sample wave, and then take the distance between each country in the sample and this set of coordinates. For example, if Wave 1 contained mostly Western countries, then the “average” geographic coordinates would be in the West, and most countries would have low geographical distinctiveness scores because they would be relatively close to these coordinates.

This measure has two advantages. First, it allows us to estimate whether the WVS is becoming more geographically diverse over time. This is what we find: geographic distinctiveness is rising over time, $b = 0.15$, $SE = 0.03$, $t = 6.06$, $p < 0.001$, 95% CIs [0.10, 0.20]. In other words, countries in the later WVS timepoints are more geographically heterogeneous than countries in the earlier WVS timepoints.

Second, we can also control for this measure of geographic distinctiveness in the critical multilevel model where value distinctiveness is regressed against timepoint (see Table 1). Controlling for this measure allows us to control for temporal changes in the heterogeneity of the WVS sample. It also controls for spatial autocorrelation in a more continuous way than continent random effects, since geographically proximal countries like Japan and China will have very similar geographic distinctiveness scores in each

wave (since they will be about the same distance from the “average” latitude and longitude coordinates).

Our key value divergence effect (i.e., the relationship between time and value distinctiveness) replicates in a regression model controlling for geographical distinctiveness, $b = 0.003$, $SE = 0.0005$, $t(9,091) = 5.96$, $p < 0.001$, 95% CIs [-0.002, 0.004]. In other words, changes over time in the geographic heterogeneity of countries does not account for value divergence. This analysis adds an important robustness check to our original approach (continent random effects), and further suggests that value divergence is not an artifact of changing WVS sample composition over time.

We now note this new analysis in our main text:

Value distinctiveness has been rising over time, $b = 0.003$, $SE = 0.0005$, $t(9,091) = 5.73$, $p < 0.001$, 95% CIs [0.002, 0.004], indicating that countries have diverged in their values over time. We also replicated this effect while controlling for spatial autocorrelation more continuously using a similar approach that we used to calculate value distinctiveness, $b = 0.003$, $SE = 0.0005$, $t(9,144) = 5.96$, $p < 0.001$, 95% CIs [-0.002, 0.004] (see Supplementary Methods for details). This continuous method further addressed the concern that our results were biased by interdependence of datapoints, often called “Galton’s problem.”

And we summarize it in greater depth in our supplemental materials (this text recapitulates many of the points that we have made in earlier in the letter):

Replicating Results Controlling for Geographic Distinctiveness. *One of the most significant challenges for our analysis is controlling for the changing composition of the WVS. Since the WVS uses distinct countries at each timepoint, it is important to determine whether longitudinal effects are real, or whether they are statistical artifacts associated with the cohort of countries changing over time. This problem is particularly pernicious because the first WVS wave included the smallest sample of countries, and these countries tended to be wealthier and more homogeneous than one would expect from a truly random cross-cultural sample.*

In our main text and supplementary materials, we summarize several measures of dealing with this problem, including (a) restricting our sample to include only countries that participated in many WVS waves, (b) replicating our key analyses at the decade level across countries which provided data in the 1990s, the 2000s, and the 2010s, and (c) separating cohort effects and longitudinal effects using centering procedures. We also control for Galton’s problem—the non-independence of nations—using continent random effects in our regression models. This is one parsimonious, albeit imperfect, way to control for non-independence because countries from the same continent are usually more similar than countries from different continents³⁷. The method is imperfect because it does not account for interdependencies within continent. For example, India, Japan, and China are all Asian countries, but China and Japan are more similar than either country is to India.

Here we summarize one additional step that we took to controlling for the changing sample characteristics of the WVS across waves: This step involved computing the “geographical distinctiveness” of each country at each WVS wave using the same method that we computed value distinctiveness. In other words, we computed the average latitudinal and longitudinal coordinate across all countries in each WVS wave, and then we computed each country’s distance from this average coordinate. For example, if Wave 1 contained mostly Western countries, then the “average” geographic coordinates would be in the West, and most countries would have low geographical distinctiveness scores because they would be relatively close to these coordinates.

This measure has two advantages. First, it allows us to estimate whether the WVS is becoming more geographically diverse over time. This is what we find: geographic distinctiveness is rising over time, $b = 0.15$, $SE = 0.03$, $t(10,440) = 6.06$, $p < 0.001$, 95% CIs [0.10, 0.20]. In other words, countries in the later WVS timepoints are more geographically heterogeneous than countries in the earlier WVS timepoints. Second, we can also control for this measure of geographic distinctiveness in the critical multilevel model where value distinctiveness is regressed against timepoint (see Table 2). Controlling for this measure allows us to control for temporal changes in the heterogeneity of the WVS sample. It also controls for spatial autocorrelation in a more continuous way than continent random effects, since geographically proximal countries like Japan and China will have very similar geographic distinctiveness scores in each wave (since they will be about the same distance from the “average” latitude and longitude coordinates).

Our key value divergence effect (i.e., the relationship between time and value distinctiveness) replicates in a regression model controlling for geographical distinctiveness, $b = 0.003$, $SE = 0.0005$, $t(9,144) = 5.96$, $p < 0.001$, 95% CIs [-0.002, 0.004]. In other words, changes over time in the geographic heterogeneity of countries does not account for value divergence. This analysis adds an important robustness check to our original approach (continent random effects), and further suggests that value divergence is not an artifact of changing WVS sample composition over time.

Reviewer #2 (Remarks to the Author):

- 8. This paper presented a series of analyses of World Values Survey data, which indicate that worldwide endorsement of cultural values have become more variable and divergent over the last 40 years. This study addresses important big questions about global changes in culture, by finding a novel way to approach the wealth of information contained within the World Values Survey. While this is not the only method that could be used to address the underlying theoretical claims, it is certainly an appropriate and interesting approach that can complement other methods (such as those that rely on other dimensions of cultural traits or more specific case studies). The data and code that they present could also be valuable for researchers who want to test alternative explanations for how countries are diverging over time, to further build on the current findings.**

The present analyses provide a nice snapshot of overall trends across a wide variety of cultural values and a comprehensive set of countries, which suggests a broad pattern that future researchers could investigate using more targeted investigations of specific issues or locations. The authors also do a good job of stress-testing the findings in several ways, such as using multiple indicators of cultural divergence, analyzing different subsets of countries, and testing the variable effects across different items, and find compellingly consistent findings across different ways of examining the data. Overall, I believe this paper would be of interest to many researchers across several disciplines in the social sciences.

The visualizations are excellent – they are quite complicated, which means they can take longer to interpret, but they also contain a lot of rich information by layering several variables into the figures, and therefore provides even more insight into the results.

Thank you for this very helpful review. We are so glad that you found the results interesting and the visualizations to be informative. We agree that our open access code could be a useful tool for future researchers. We have worked hard to extensively annotate our code so that researchers can use it to test different questions, such as why some values are more heterogeneous than others across cultures. As you mention, this code could also be the basis for future researchers who hope to propose novel explanations of value divergence. The primary goal of this paper is to document value divergence. We propose some possible explanations throughout the paper, but we view these explanations as tentative, and we are eager to see future papers which extend our methodology to provide more in-depth accounts of value divergence.

In our revision, we have made several key changes to address your points of feedback. For example, we have provided more clarity about the relationship

between value variation and value distinctiveness, and we have revised the figures and figure captions so that they are easier to interpret.

9. The two measures that they use to indicate cultural divergence – “value variation” and “value distinctiveness” – are both plausible ways to index the strength of divergences across different countries. The authors do a good job explaining conceptually how these measures are calculated in the introduction, so that readers can understand the results, while providing more details in the methods and supplemental materials which would allow interested researchers to replicate their analyses. Because these metrics are not typical in pre-existing literature, it was a bit hard at first to parse how these metrics were distinct from one another, and in the results I was sometimes confused about which metric was being referred to in each analysis or figure. For example, in Figure 1 the value is simply labeled “Divergence” on the x-axis, but I think this is derived from value variation, which is something that could be explained in the figure caption or on the figure itself. I also found it sometimes unclear what the predictor or outcome variables were in the regression analyses described briefly in the results. Adding sub-headings in the results section (if allowed by journal style guides) could also help distinguish when analyses shift from focusing on one indicator of cultural divergence to the other. The authors already do a decent job at explaining their measures and analyses, but a little more hand holding could help readers, especially in the results section. Their breaks in the results to summarize the overall pattern and what it might mean are helpful to keep the reader on track, but being even more specific in the figure caption or regression descriptions could be helpful.

Thank you for these suggestions. We agree that results should be as clear as possible when using new variables.

In response #5 to reviewer 1, we summarize several steps that we have taken to make our results as clear as possible. These involve moving the table which defines value variation and value distinctiveness to the introduction. Following reviewer 3’s recommendation, we have now explicitly stated that these measures are conceptually the same, but they focus on different levels of analysis. Value variation focuses on the level of the item (the standard deviation of each item across countries), and value distinctiveness focuses on the level of the country (the distance of each country from the global average across values).

We have also made several changes which are directly inspired by your comment:

- We have revised the axis label of Figure 1. As you point out, the label “divergence” could be confusing to readers, and so we have replaced it with “Change in Value Variation,” where more positive values represent a rise in value

variation over time (value divergence) and more negative values represent a decline in value variation over time (value convergence).

- We have added sub-headings to our results section, which is permitted by the journal. Our analyses of value variation fall within the subsection “value divergence at the item level” and the analyses of value distinctiveness fall within the subsection “value divergence at the country level.”

Thank you again for these recommendations.

10. In regression equations predicting country-level value distinctiveness from a variety of plausible variables (e.g., inequality, globalization, political rights), GDP was consistently a strong predictor, and often only the significant one. However, I wonder how strongly GDP is correlated with other predictors in the model, and therefore might be artificially sucking up variance from other predictors (e.g., rich countries might also be more globally connected and have more political rights).

This is an interesting perspective. We agree that it is possible that other predictors could be statistically significant without controlling for GDP per capita, but we do not see the variance explained by GDP in this model as “artificial”: it seems to be a very real association between wealth and value distinctiveness. If a variable like globalization was only significantly linked with value distinctiveness without controlling for GDP per capita, then it would suggest that the relationship between globalization and value distinctiveness is spurious, because it is confounded with GDP: wealthy countries also are more likely to have high globalization scores, which is the only reason why globalization might be linked to value distinctiveness.

Nevertheless, we agree that it is interesting to consider how the effects would change without controlling for GDP per capita, so we have added that model to the supplemental materials in a sub-section named “Replicating Analyses of Value Distinctiveness Without GDP per Capita.” Interestingly, we find that none of the other fixed effects are significant other than the effect of time, which suggests that wealth is the only variable in our cross-cultural sample which is reliably associated with value distinctiveness. We have pasted this analysis below (and juxtapositioned the relevant text and table together for ease of interpretation):

GDP per capita was the only significant fixed effect in Table 2 of our main text, whereas globalization, inequality, political rights, and distance from equator were not significantly associated with value distinctiveness. In a follow-up analysis, we tested for how results changed when we removed GDP per capita from the model. Since GDP per capita is correlated with higher levels of globalization ($r = 0.65$), lower levels of inequality ($r = -0.29$), higher levels of political rights ($r = 0.49$), and greater distance from the equator ($r = 0.16$), we reasoned that it may have accounted for the non-significant associations involving these fixed effects in our main analyses. However, this did not seem to be the

case. None of the other covariates reached significance in a model which did not include GDP per capita. We report these results in Supplementary Table 8.

Supplementary Table 8.
Associations with Value Distinctiveness in Multiple Regression Without GDP Per Capita

	b (SE)	t value	df	p value	95% CIs
Timepoint (C)	-0.001 (0.004)	-0.29	74.48	0.77	-0.01, 0.01
Timepoint (L)	0.002 (0.001)	2.19	858.43	0.03	0.0002, 0.004
Political Rights	-0.005 (0.01)	-0.76	1980.97	0.45	-0.02, 0.01
Gini	0.02 (0.02)	1.02	151.62	0.31	-0.02, 0.05
Distance from Equator	-0.01 (0.01)	-0.54	56.44	0.59	-0.03, 0.02
Globalization	0.02 (0.02)	1.29	399.83	0.20	-0.01, 0.05

11. The authors cluster the data into a 2-dimensional structure using PCA, which they then use to calculate the distance between countries across many different variables. Previous authors have also tried to provide a summary of the underlying structure across many WVS items (e.g., the Inglehart-Welzel cultural map of Traditional values vs. Secular-rational values and Survival values vs. self-expression values). Are the presently-identified dimensions comparable to the underlying value dimensions identified by previous researchers? Discussing this could help draw clearer connections between the present paper and prior research.

This is a great idea! We have now added a section where we correlate both PCs with Welzel's emancipative and secular value dimensions. These dimensions are useful because we use them earlier in the paper to show that value divergence is highest for emancipative values. We also correlate the PCs with Inglehart's post-materialist index. We summarize these analyses on page 12 of the main text:

In our final analysis, we explored which countries hold similar values to one another. We first fit a principal components analysis (PCA) to detect underlying dimensions in how the 40 values varied across nations. Two principal components (PCs) explained 80.70% of variation (see Materials and Methods). To understand the meaning of these dimensions, we regressed each dimension against previously developed dimensions of value variation: Welzel's secular values and emancipative values, and Inglehart's 12-item post-materialist index. PC1 was strongly positively linked to emancipative values, $b = 0.76$, $SE = 0.04$, $t(214.70) = 19.07$, $p < 0.001$, 95% CIs [0.67, 0.84], and also positively linked to secular values, $b = 0.27$, $SE = 0.03$, $t(213.90) = 8.73$, $p < 0.001$, 95% CIs [0.21, 0.33], whereas PC2 was negatively related to secular values, $b = -0.16$, $SE = 0.07$, $t(206.98) = -2.41$, $p = 0.02$, 95% CIs [-0.30, -0.02] (see Supplementary Methods for more information about these models).

We used these PCs as coordinates in a two-dimensional value space in which countries with similar values are closer together than nations with different values (see Figure 5). We next fit regression models for each timepoint to test whether geopolitical characteristics could explain which countries were closer together in value space and which countries were further apart.

We also present the analyses in more depth within our supplemental materials:

Correlating PCs with Dimensions of Values. In our main text, we summarize analyses in which we correlated PC1 and PC2 from our PCA with previously established dimensions of values. We chose three dimensions: Welzel's secular and emancipative value dimensions (indexed as Y010 and Y010, respectively, in the WVS), and Inglehart's 12-item post-materialist index (indexed as Y001 in the WVS). Since these indices are stored as variables in the longitudinal WVS file, they were easy to retrieve and to correlate with the PCs.

It is important to note that these dimensions are not independent of each other. Supplementary Table 14 shows that the dimensions are correlated quite robustly. The coefficients in this table represent standardized estimates from cross-classified multi-level models with country-wave means nested in countries and waves. Because of these covariances, we chose to model the dimensions together as fixed effects in a multiple regression so that we could estimate the distinct contribution of each dimension to explaining variance in the PCs.

Supplementary Table 14.			
Associations Between Value Dimensions			
	Secular	Emancipative	Post-Materialist
Secular	-		
Emancipative	0.56**	-	
Post-Materialist	0.13*	0.63**	-

Note. Single-starred coefficients are significant at $p < .05$. Double-starred coefficients are significant at $p < .001$.

Supplementary Table 15 summarizes the coefficients from models where each PC was regressed on these three value dimensions. The models were cross-classified multi-level models with country-wave means nested in countries and waves. PC1 was strongly positively linked to emancipative values and also positively linked to secular values, whereas PC2 was inversely related to secular values, but not to the other dimensions. Neither PC was reliably associated with post-materialist values above and beyond the Welzel dimensions.

Supplementary Table 15.					
Associations Between Principal Components and Previously Established Value Dimensions					
	b (SE)	t value	df	p value	95% CIs
PC1					
Secular	0.27 (0.03)	8.73	231.90	< 0.001	0.21, 0.32
Emancipative	0.76 (0.04)	19.07	214.70	< 0.001	0.67, 0.84
Post-Materialist	-0.06 (0.03)	-1.98	246.50	0.05	-0.12, -0.01
PC2					
Secular	-0.16 (0.07)	-2.41	206.98	0.02	-0.30, -0.02
Emancipative	0.16 (0.09)	1.85	144.10	0.07	-0.01, 0.34
Post-Materialist	0.01 (0.06)	0.01	217.10	0.92	-0.12, 0.13

Reviewer #3 (Remarks to the Author):

12. This is an excellent paper on a very interesting topic and an important contribution to the behavioral and social sciences. The authors of the paper are asking a simple and important question – is the world converging or diverging in cultural values? The question is simple but answering it conclusively is no easy task. The authors have done an excellent job of drawing on multiple waves of data from the World Values Survey, using the most advanced statistical methods, and conducting several robustness checks. The paper is written for a broad audience.

The authors have done a good job of ruling out the biggest counter explanation, that these trends are artifacts of the WVS sampling getting more diverse over time. They rule this out in several ways. This is a big strength of the study.

The authors do a very good job providing context on the different theories that offer competing predictions regarding the divergence or convergence of worldwide values. Below are suggestions, questions, and recommendations, with the goal of strengthening the paper’s contribution and potential impact.

Thank you for the encouraging feedback, and for the excellent suggestions throughout your review. As you can see, we have incorporated all of these suggestions into our revised manuscript. We hope that you find the revision to be an improvement.

13. Given the role “emancipative values” have in the results, it might be worth it to break down emancipative values and Welzel’s theory in greater detail. It was not entirely clear what an emancipative value was. For example, some research showing there has been a “global rise of individualism” is discussed alongside a study that showed “emancipative values are diffusing around the world” but the distinction between the two is not clear to me. I think this problem is present throughout the manuscript, given how much emancipative values are discussed, but a sharper conceptual analysis and discussion could deal with this problem.

We apologize for the confusion about the term “emancipative”; this term is not particularly common in anthropology or psychology, and we should have been clearer about its meaning in the original submission.

Welzel, who defined emancipative values and developed the measure that we use, viewed emancipative values as closely linked with individualism. Here is a quote from his paper “Generalized Trust: The Benign Force of Emancipation” (Welzel & Delhey,

2015, *JCCP*) where he summarizes the relationship (the citation here is to his book, *Freedom Rising*):

“Individualist-versus-collectivist legacies are linked with the emancipatory tendencies of human empowerment, visible in a strong positive correlation between emancipative values and individualism, and an equally strong negative correlation between collectivism and emancipative values (Welzel, 2013, pp. 82-84). This linkage suggests that emancipation and empowerment are inherent features of individualism.”

Another way of saying this is that emancipative values are values that promote self-expression and individual autonomy (hallmarks of individualism) over obedience to the in-group. We have rephrased the definition of emancipative values in our revision’s introduction, and drawn a direct comparison to collectivism to make this link clearer to readers (we have bolded relevant text in the excerpt):

*New theoretical models emerged during this time, with varying levels of overlap with the older theories. Inglehart’s postmaterialist thesis suggested that globalization alone does not lead to cultural convergence, but that economic development in particular induces a shift from values prioritizing group obedience to values prioritizing self-expression and individual autonomy^{9,18}. Welzel labeled these latter values as “emancipative,” and proposed a “human empowerment” sequence in which wealth and security encourage cultures to espouse more emancipatory values, which in turn foster participatory democracy⁸. **Emancipative values bear a conceptual overlap and statistical correlation with individualizing values, insofar as both sets of values emphasize the autonomy and needs of the individual over those of the group⁸.** The post-materialist thesis therefore reproduces the assumption that cultural convergence will inherently be in the direction of Western individualism, but adds the caveat that this convergence requires economic prosperity and financial security.*

14. It would be helpful to clarify how the authors are conceptualizing the relationship between the “value distinctiveness” and “value variation” measures. Are these two measures assessing the exact same construct but in different ways? In this sense, is the value distinctiveness measure a robustness check on the results using the “value variation” measure, or are these two measures measuring subtly different aspects of the construct? My impression was that these are two different conceptualizations of the same construct, and that “value distinctiveness” is essentially a robustness check for the “value variation” measure, but more clarity would be useful and important.

Thank you for the encouragement to be clearer about these measures. Your comment echoes suggestions by reviewers 1 (see response #5) and 2 (see response #9). You are correct that these two conceptualizations represent the same construct, but at different levels of analysis. Value variation focuses on the level of the item (the standard deviation of each item across countries), and value distinctiveness focuses on the level of the country (the distance of each country from the global average across values).

Based on your encouragement, we now explicitly write in our introduction:

These measures are meant to represent the same outcome—convergence vs. divergence of values across world nations. However, they vary in their level of analysis. The first measure, “value variation,” focuses on divergence at the level of the WVS item whereas the second measure, “value distinctiveness,” focuses on divergence at the country level.

15. Complementing on these two measures of calculating divergence, there is a third method that could be used to significantly bolster the paper’s robustness checks. Get cultural distances by calculating average CF_{ST} of all countries per wave, then find out if cultural distances are growing over time, which would indicate divergence. See Muthukrishna et al., 2020 for this method.

M. Muthukrishna et al., Beyond Western, Educated, Industrial, Rich, and Democratic (WEIRD) psychology: Measuring and mapping scales of cultural and psychological distance. *Psychological Science*, 31, 678–701 (2020).

Thank you for this suggestion! We think it is a great idea, and we have incorporated the analysis into the supplementary materials of our revision. Our findings using the CF_{ST} approach are highly similar to the findings using our value variation and value distinctiveness measures, which is not surprising since the measures are all assessing the same thing. This is a useful robustness check and we are grateful for the opportunity to include it. We have pasted the new section below:

***Replicating Results with Cultural Fixation Indices.** In addition to using our indices of value variation and value distinctiveness, we also found evidence for value divergence using Cultural Fixation Indices (CF_{ST}). The F_{ST} metric was first developed in genetics to measure how genotype frequencies for each subpopulation differ from expectations assuming random mating. The statistic became popular because it is easy to interpret as a measure of general ratio of between-group to total variance. An F_{ST} near 0 indicates that individuals between populations are about as different as individuals within populations, whereas an F_{ST} near 1 indicates that all variance between individuals exists between populations.*

In an influential 2020 paper, Muthukrishna and colleagues⁴ developed a CF_{ST} metric which they applied to the items of the WVS to quantify cultural distance between populations. They published metrics of cultural distance in this paper, but also equations that researchers could use to compute cultural distance across subsets of categorical and continuous items, or even for new datasets. We translated their equations into R code, and computed CF_{ST} matrices for each WVS wave using the same sampling and normalizing criteria that we used for our main analyses. In particular, we computed cultural distance using the 40 items displayed in Figure 1, among all countries included in at least two waves of the WVS. Our R code for computing cultural distance is publicly

available at <https://osf.io/f9bz7/>. In our approach, binary items or 3-level items were treated as categorical and CF_{ST} was computed using the categorical equations, whereas items with 4 or more levels were treated as continuous and CF_{ST} was computed using the continuous equations. The estimates of these analyses were sensible. For example, the lowest CF_{ST} estimate was between New Zealand and Australia in wave 7 ($CF_{ST} = 0.008$), whereas the highest CF_{ST} estimates was between Sweden and Bangladesh in wave 4 ($CF_{ST} = 0.52$). The highest CF_{ST} score featuring two countries from the same continent was between Japan and Iraq ($CF_{ST} = 0.50$).

After computing these CF_{ST} scores, we then estimated whether the mean cultural distance between countries has increased over time, which would be supportive of value divergence. We conducted this analysis by melting the CF_{ST} matrix from each wave into a long dataframe of pairwise country comparisons, and then binding together the wave-specific dataframes by row. Next, we fit a cross-classified model with observations nested in the first and second countries in the pairwise comparison. In this model, CF_{ST} value was regressed on timepoint. The fixed effect of timepoint was significant and positive, $b = 0.004$, $SE = 0.0006$, $t(6,119) = 7.29$, $p < 0.001$, 95% CIs [0.003, 0.005]. The effect remained significant and positive after further nesting estimates within the continents associated with each country in the pairwise comparison, $b = 0.004$, $SE = 0.0006$, $t(6,097) = 7.16$, $p < 0.001$, 95% CIs [0.003, 0.005]. These effects were consistent with our main text findings, and offer further support for worldwide value divergence. Value divergence using this CF_{ST} approach also showed non-linear growth. When we fit a second-order polynomial model, we found a significant and positive linear effect of timepoint accompanied by a significant and negative quadratic effect. Supplementary Table 7 reports the output of this model.

Supplementary Table 7.

Second-Order Polynomial Model of CF_{ST} Over Time

	b (SE)	t value	df	p value	95% CIs
Timepoint (Linear)	0.02 (.003)	7.42	6,045	< 0.001	.02, .03
Timepoint (Quadratic)	-0.002 (.0003)	-6.24	6,045	< 0.001	-0.003, -0.001

In Supplementary Figure 5, we illustrate the average CF_{ST} scores of each continent—contrasted with countries from other continents—over time. This figure shows that every continent became more culturally distant from other continents from 1981 to 2022. For example, the average European country became more culturally different from the average non-European country.

Supplementary Figure 5. Continents by their mean CF_{ST} score (contrasted with countries in other continents) at each WVS wave. Mean CF_{ST} has risen for all continents over time, which is consistent with value divergence.

16. The results section was relatively clear, and the figures and tables the authors chose to include in the main text are well done. The discussion is also very well written and contextualizes many of the findings. The GDP per capita analyses require a little more context. For example, what is the statistical power for the analyses showing that “the effect of GDP per capita was significantly weaker in Africa and Asia compared to the effect in Europe” or that “wealth was not associated with value distinctiveness in African and Asian countries”. The WVS is a massive dataset, and these analyses utilized MLM’s, so it’s entirely possible they are well-powered, but given that it appears these comparisons are occurring at the country-level, some discussion of statistical power is necessary. This would help readers determine how much stock they should put into these subset analyses, given that there doesn’t appear to be a good explanation for why this is the case.

Thank you for the encouragement, and for your suggestion. We agree that the analyses could use some more context, including notes about the sample size of countries

involved and the observed power of the key interactions. We have revised the paragraph to include these details.

After running these new analyses, we agree that the analyses must be taken with caution because the power is somewhat low (~70% and ~80%), respectively. However, the difference in the effect of GDP per capita across European countries and Asia/African countries is so stark that we feel that this interaction is worth reporting. To this point, the wealthiest country in Asia, Singapore, has the least distinct values of all Asian countries in the 7th WVS wave. We decided that the best approach would be to leave the analyses in the main text of the power, but report the power transparently and note that the readers should interpret these two-continent contrasts with more caution than tests which included the full sample of countries in our sample.

This is the revised paragraph in our main text:

Further analyses found that the association between wealth and value distinctiveness varied across world region. Wealth was associated with value distinctiveness among European countries, $b = 0.07$, $SE = 0.02$, $t(16.92) = 3.11$, $p = 0.003$, 95% CIs [0.03, 0.11]. However, in a regression model where we interacted GDP per capita with continent dummy-codes, we found that the effect of GDP per capita was significantly weaker in Asia, $b = -0.11$, $SE = 0.04$, $t(15.71) = -2.83$, $p = 0.01$, 95% CIs [-0.17, -0.04], and Africa, $b = -0.24$, $SE = 0.10$, $t(171.10) = -2.36$, $p = 0.02$, 95% CIs [-0.43, -0.04], compared to Europe. These multilevel model estimates were based on smaller subsets of countries, with 48 countries in the Europe vs. Asia comparison (simulated power = 82.80%) and 36 countries in the Europe vs. Africa comparison (simulated power = 70.00%), so readers may interpret these findings with more caution than tests that included the full cross-cultural sample. But it is also worth noting that the association between GDP per capita and value distinctiveness was not only smaller in these regions than in Europe but non-significant in both Africa, $b = -0.17$, $SE = 0.10$, $t(105.70) = -1.69$, $p = 0.09$, 95% CIs [-0.35, 0.03], and Asia, $b = -0.04$, $SE = 0.03$, $t(15.48) = -1.14$, $p = 0.27$, 95% CIs [-0.10, 0.03]. Supplementary Table 13 summarizes the complete set of coefficients for these models. Our Supplementary Methods provide more details about the power analysis simulations.

As you point out, the power of a multi-level model is more complicated to compute than the power of an OLS regression, since variance can be decomposed across multiple samples (e.g., of countries, of variables, or of continents). For this reason, we provide more details and context for our power analyses in our supplemental materials:

In our main text, we also report power estimates for the key interactions contrasts of Europe vs. Africa (observed power = 70.00%) and Europe vs. Asia (observed power = 82.80%). Power is difficult to parse in a mixed-effects model. For example, degrees of freedom for the residuals in a standard OLS regression is simply the difference between the sample size and the number of parameters being estimated, including the intercept. In contrast, mixed effects models introduce random effects that account for variations in the data that aren't captured by fixed effects alone. This makes the calculation of degrees of freedom more complicated. Satterthwaite's approximation is one method to

estimate the degrees of freedom in mixed models, and it is the default approach in the lmer models that we fit. The Satterthwaite method calculates degrees of freedom based on the variance components of the model, essentially approximating the distribution of the test statistic by considering the ratio of variances. It provides a way to compute the appropriate degrees for F-tests in the presence of the additional complexity from the random effects. However, it means that power is often not easy to infer by eye because the degrees of freedom are not an intuitive indicator of sample size, nor is there one sample size to gauge power, since variance can be decomposed across samples of countries, continents, or variables. For these reasons, standard power calculators are often inappropriate for mixed effects models.

One approach to overcome this limitation is to use simulation. For our observed power estimates, we used “simr,” which is a package that can accept mixed effects models as arguments, and then simulate the model n times ($n = 500$ in our simulations) to detect the observed power of a given fixed effect given the structure of the model. This approach is considered the gold standard for simulating power in mixed effects models³⁹, but it is still flawed, especially when applied post-hoc because there is no guarantee that the observed features of the sample data—such as the effect size of a key fixed effect—represents the true nature of the population. This risk is not as dire in our model, because our sample of countries *does* represent a large share of all the world’s countries, but we still encourage readers to treat the exact power estimates with caution.

Finally, we would like to respond to your point that “there doesn’t seem to be a good explanation for why this is the case.” We do think that there is some precedence for this finding in the literature, and so we have tried to provide a better summary of this perspective, which helps explain how rising wealth could lead to divergence between countries.

These findings suggest a provocative possibility: rises in global wealth may be responsible for the worldwide divergence of values. Most countries have become wealthier over time—Western countries but also many non-Western countries like Singapore, Hong Kong, and South Korea⁵¹. This means that most countries were poorer in the earlier (vs. later) waves of the WVS. Western nations in these early waves held more emancipative values than non-Western nations, but not by a large degree. As time passed, rising wealth led Western countries to adopt more emancipative values, whereas wealth had much less of an impact on the emancipative values of Asian or African countries. These trends led wealthy Western nations to hold increasingly more distinct values over time.

Consider Hong Kong and Canada, where wealth has followed a remarkably similar trend, but values have diverged. Both countries had a GDP per capita of approximately \$25,000 in 2000, which doubled to approximately \$50,000 by 2020^{52,53}. Over the same time interval, beliefs that homosexuality was justifiable rose in Canada from 0.49/1.00 to 0.74/1.00. Perceived justifiability of homosexuality also rose in Hong Kong, but only from 0.29/1.00 to 0.44/1.00. This means that the gap in means between Hong Kong and Canada increased by 50% during this period. At the same time, one of the fastest-changing values in Hong Kong was belief in children’s work ethic. From 2000 to 2021,

belief that responsibility was an important value in children rose from 0.26/1.00 to 0.80/1.00 whereas it fell in Canada from 0.72/1.00 to 0.50/1.00.

This example shows that wealth can sometimes lead to counter-intuitive effects on values and produce divergence rather than convergence. This finding is consistent with Eisenstadt's "multiple modernities" thesis that countries follow their own trajectories of modernization¹⁹. It is also consistent with Huntington's observations that rising wealth and influence in East Asia led to a re-affirmation of traditional Confucian values²². However, it is important to acknowledge that factors such as migration, political change, and decolonialization could also contribute to these trends. For example, sovereignty over Hong Kong transitioned from the United Kingdom to China in 1997, which may have affected people's values in the region. More research is necessary to determine the causal relationship, if any, between wealthy and value change.

To make our explanation clearer, we used your advice to contextualize the trends by using examples of real countries. We also note that our interpretation is still speculative, and future research using more causal modeling might be better able to identify mechanisms.

17. The comparison of Australians' changing opinions on childhood obedience compared to the changing opinions of Indians was very helpful in providing some texture to some of the effects. Is the growing divergence driven by some countries remaining constant in their values over time, while Western "progressive" countries are undergoing change? This example suggests that it is countries going in opposite directions. A few more examples of this kind for some of the highest changing values would be helpful. A table of some sort providing a few more example items and changes in responses from Wave 1 to Wave 7 for prototypical countries would go a long way to contextualizing these trends.

We are glad you found the example helpful. Based on your feedback, we added a similar example when we contrast value change in wealthy Asian countries vs. wealthy European countries (our previous response).

You raise an interesting point about whether divergence is driven by countries moving in opposite directions vs. some countries changing whereas others remain stable. Both cases are true depending on the items and countries involved. Figure 2 shows that, when averaging across the 7 "emancipative" items, divergence is driven by Western countries changing whereas African and Asian countries remaining mostly the same. But there is a lot of heterogeneity beneath this aggregate plot. Cases like the Australia vs. India example show that the values of some countries are moving in opposite directions.

We have tried to clarify this heterogeneity in our revision (bolded text added for emphasis here):

Rises in value variation for the 7 most divergent items are displayed in Panel A of Figure 2. The clearest rises in value variation come from timepoints 1–5 and plateau across timepoints 5–7. Panel B of Figure 2 illustrates changes over time in the means of these 7 values, which are aggregated to the continent level and coded such that higher values mean more emancipative values. This plot shows that Oceanic, European, North American, and South American countries have progressively endorsed more emancipative values, whereas endorsement of these values has been stable across Asian and African countries. **Divergence on some values is driven by countries moving in opposite directions, as in the case of Australia vs. India (see Supplementary Table 16 for other examples). Other values diverged because their mean rate of endorsement changed in some countries but remained the same in others.** We provide detailed methodological details about these items and countries in our Supplementary Information (e.g., Supplementary Tables 2-5).

We have also added Supplementary Table 16, which provides more examples of pairs of countries with diverging values, to complement the Australia vs. India example. We have tried to include a range of countries in this example.

Supplementary Table 16.		
Examples of Diverging Values and Notable Countries		
Item Label	% Change in Variation	Example of Diverging Countries
Important Child Qualities: Obedience	+54.27%	Dropped in Moldova from 0.39/1.00 to 0.17/1.00 Rose in Pakistan from 0.32/1.00 to 0.49/1.00
Justifiable: Divorce	+91.57%	Dropped in Kyrgyzstan from 0.26/1.00 to 0.14/1.00 Rose in Chile from 0.28/1.00 to 0.57/1.00
Important Child Qualities: Religious Faith	+97.49%	Dropped in Japan from 0.06/1.00 to 0.04/1.00 Rose in Armenia from 0.12/1.00 to 0.34/1.00
Neighbors: Immigrants, /Foreign Workers (Item Reflects Aversion to Having as Neighbor)	+298.21%	Dropped in Indonesia from 0.40/1.00 to 0.17/1.00 Rose in Iran from 0.10/1.00 to 0.42/1.00
Justifiable: Euthanasia	+34.85%	Dropped in Turkey from 0.22/1.00 to 0.17/1.00 Rose in Spain from 0.39/1.00 to 0.51/1.00

Note. The “% Change in Variation” variable is calculated by taking the absolute difference of the standard deviation of country means between timepoint 1 and timepoint 7, and then dividing it

by the standard deviation at timepoint 1. In this equation, 100% change represents an item where variation across countries has doubled over time.

18.p. 8 “This pattern of results is consistent with the illustration in Figure 2 that values have diverged non-linearly. These analyses address the concern that value divergence is an artifact of changing country membership in the WVS over time; value divergence is visible even when the sample of countries is held constant over time.”

What is the tipping point? It was not explained and it’s not clear from the figure (there seem to be multiple inflection points?)

If we understand correctly, you are asking about the nature of non-linear value divergence. Was there a specific inflection point where value divergence plateaued? This is a very good question.

This “tipping point” seems to look different for different items (e.g., in Figure 2, variation appears to stop rising for some items after wave 4, for others after wave 5, and for others after wave 6). But we appreciate that it would be useful to identify whether the functional form of value divergence has a general inflection point, or whether value divergence has gradually slowed over time (i.e., values diverged more between 1980 vs. 1990 than between 1990 vs. 2000, and values diverged more between 1990 vs. 2000 than between 2000 vs. 2010, etc.).

In our revision, we have now added new analyses that evaluate different functional forms for value divergence. These new analyses evaluate the probability that non-linear value divergence decelerated at a specific “tipping point” or whether it has gradually decelerated at a constant rate. We evaluated this possibility with a series of non-linear models. We fit these models using both of our outcome variables: value variation and value distinctiveness. We fit five models for each outcome variable: A linear model, a second-order polynomial (quadratic) model, and spline models with discontinuities at timepoint 3, timepoint 4, and timepoint 5. Spline models are also known as piecewise polynomial regressions, and they are characterized by one or more discontinuities in a functional form.

In our main text, we have revised the key paragraph that you pointed out to report key results from this new analysis:

This decade-based analysis provides evidence for value divergence even when we fix the country sample over time. It also suggests that value divergence has been non-linear. In our Supplementary Methods we evaluate different non-linear models of value divergence. These models suggest that the pace of value divergence has gradually slowed over time, rather than subsiding at a particular point.

And we report a more thorough set of analyses and interpretation in the supplementary materials:

Non-Linear Value Divergence. *In our main text, we noted that value divergence has had a non-linear functional form. Values seem to have diverged most sharply in the 1980s and 1990s, and then diverged more gradually in the 2000s and 2010s. Readers may wonder whether this non-linear form had a single “tipping point” where value divergence slowed down, or whether this deceleration was gradual. We evaluated this possibility with a series of non-linear models. We fit these models using both of our outcome variables: value variation and value distinctiveness.*

Our parameterization of these non-linear models was identical to the models we report in the main text. They were multilevel models. As in the main text, the regressions involving value distinctiveness were nested within value, country, and continent, whereas the regressions involving value variation were nested within value (because the data had already been aggregated across country). We fit five models for each outcome variable: A linear model, a second-order polynomial (quadratic) model, and spline models with discontinuities at timepoint 3, timepoint 4, and timepoint 5. Spline models are also known as piecewise polynomial regressions, and they are characterized by one or more discontinuities in a functional form. We only fit models with a single discontinuity because we only had seven total timepoints—models with multiple discontinuities are characterized by more timepoints. For each model, we extracted both AIC and BIC fit to evaluate which model provided the best fit to the data. The full range of fit coefficients are given in Supplementary Table 10.

Supplementary Table 10.		
Fit Statistics Associated with Non-Linear Models		
Model	AIC	BIC
Value Variation (Linear)	-1142.54	-1128.00
Value Variation (Quadratic)	-1160.92	-1142.75
Value Variation (Spline_a)	-1154.92	-1136.57
Value Variation (Spline_b)	-1154.97	-1136.57
Value Variation (Spline_c)	-1154.75	-1136.57
Value Distinctiveness (Linear)	-21898.86	-21855.23
Value Distinctiveness (Quadratic)	-21930.96	-21880.05
Value Distinctiveness (Spline_a)	-21919.65	-21868.75
Value Distinctiveness (Spline_b)	-21918.97	-21868.07
Value Distinctiveness (Spline_c)	-21917.74	-21866.84

Note. The best-fitting model for each outcome variable is bolded here.

For both value variation and value distinctiveness, the quadratic model had a better fit than either the linear or either of the spline models. We report the results of this quadratic model in the main text for value distinctiveness. Supplementary Table 11 summarizes the results of the model for both value distinctiveness and value variation. For both models, we found robust positive linear effects and negative quadratic effects. This suggests that countries’ values have diverged over time, and the rate of this divergence has gradually decelerated rather than changing suddenly at a single

inflection point. We also report the results of the value distinctiveness model in the main text.

Supplementary Table 11.
Results of Non-Linear Models

	b (SE)	t value	df	p value	95% CIs
Value Variation					
Linear Slope	0.14 (0.02)	6.09	238	< 0.001	0.10, 0.19
Quadratic Slope	-0.10 (0.02)	-4.44	238	< 0.001	-0.15, -0.06
Value Distinctiveness					
Linear Slope	0.56 (0.10)	5.82	9,222	< 0.001	0.38, 0.75
Quadratic Slope	-0.46 (0.09)	-5.17	10,560	< 0.001	-0.64, -0.29

We hope that these new sections address your suggestion to provide more clarity.

19.p. 11 “We found that people within countries are not diverging in values as much as countries are diverging from each other”

Does this mean nation-states are becoming more significant cultural forces over time, not less? (That is quite the opposite of the idea that the nation-state concept is being torn apart by tribalism on one end, and globalization on the other).

Based on our analyses, we cannot make any major claims about changes in within-country heterogeneity over time. This is the relevant section from our supplementary materials:

***Supplemental Information About Within-Country Heterogeneity.** We calculated within-country heterogeneity by estimating value variation across people within countries. This followed a similar approach to calculating value variation across countries. We first normalized item responses using min-max normalization, and then we took the standard deviation of responses across participants within each country (rather than across country means).*

We fit linear models to test whether within-country heterogeneity is rising within countries, which would indicate within-country value divergence. However, the evidence from these models was inconclusive. When did find evidence of within-country value divergence in a mixed effects regression where within-country heterogeneity was regressed on timepoint, with random effects for item, country, and continent, $b = 0.001$, $SE = 0.0004$, $t(9,699) = 3.25$, $p = 0.001$, $95\% CIs [0.0005, 0.002]$. However, the effect size of this model was roughly a quarter of the effect size of our between-nation value divergence model, and the effect did not reach significance when we allowed slopes to vary across items ($p = 0.15$). Similarly, when we aggregated across items and examined the trend of within-country heterogeneity within each nation, we found a highly diverse set of effects. Values heterogeneity has been rising in countries like Algeria ($r = 0.54$), Albania ($r = 0.50$), and Iran ($r = 0.39$), but declining in countries like Lebanon ($r = -0.51$), Ghana ($r = -0.44$), and the Netherlands ($r = -0.40$). The median within-country

heterogeneity trend was close to 0 (0.01), and was not significantly different from 0, $t(75) = 0.57$, $p = 0.58$, 95% CIs [-0.04, 0.07]. Because of these mixed results, we do not make any claims about whether within-country heterogeneity is rising or falling across nations over time.

We also cannot make any claims that countries are being “torn apart” by either globalization or tribalism. We make no moral claims in this paper about whether value divergence or convergence is a good or bad thing. We only provide evidence that countries have diverged in their values over the last 40 years, and that this effect seems strongest for countries which are geographically distant and have dissimilar GDP per capita and religious history (e.g., India vs. Canada).

We do see evidence that values are converging among countries in the same region (e.g., Iran and Saudi Arabia, or Japan and China). We point this out several times, including in our abstract when we write “Over time, however, geographic proximity has emerged as an increasingly strong correlate of value similarity between countries, indicating that values have diverged globally but converged regionally.”

20.p. 14. “Religion has also emerged as a robust predictor of value similarity. Countries with more similar religious profiles also have more similar values, even controlling for their similarity in wealth, geographic position, and other geopolitical features.”

This is good to know and it is a replication of White, Muthukrishna, & Norenzayan, 2021, PNAS, that is not cited or mentioned. This effect of religion on cultural values remains holding constant wealth and country.

Thank you for suggesting this paper. We agree that our finding conceptually replicates the main finding in that paper, and we have added a citation to the paper on page 14 when we report that countries with similar religious profiles show similar values.

21. Also p. 14. “Value divergence appears to be strongest for values related to tolerance and openness and may be explained by rising differences between wealthy Western countries and the rest of the world. Worldwide value divergence has been accompanied by regional convergence, with geographically proximal countries adopting more similar values over time.”

The findings in this paper suggest that Western countries, already extreme outliers to begin with, are even more so now than they were a few decades ago. Does this not suggest that the WEIRD problem has become more acute over time? This has deep implications for those behavioral and social science fields that overwhelmingly skew WEIRD in their samples. Are social scientists studying cultural bubbles that are becoming even more isolated from worldwide patterns?

This is an excellent point. We should have highlighted the WEIRD problem more prominently in the general discussion of our original submission. We agree with your inference, and we have rewritten the final paragraph of our paper to point out this problem as one of the major implications of our findings:

Our research also underscores the limitations of studying Westerners to make claims about human psychology writ large. Cross-cultural scholars have pointed out that people from Western, Educated, Industrialized, Rich, and Democratic (WEIRD) countries have psychological traits which differ from the rest of the world. This peculiarity presents an external validity problem for studies that recruit mostly WEIRD subjects, and it also presents an intellectual problem for cultural evolutionists who hope to explain this regional variation. We show that this problem has become more acute in the last forty years. WEIRD subjects have become even more peculiar. This shift makes it more crucial than ever that behavioral scientists develop their theories using data from globally representative samples.

22. In the paragraph on page 15 discussing the practical implications of value divergence for “political polarization and conflict across world countries”, it was unclear why right-wing populism is a trend consistent with the findings. Aren’t most of the right wing populism movements occurring in wealthy Western nations over the past ~10 years? If anything, wouldn’t this suggest that wealthy Western nations may be backsliding into more traditional values consistent with the rest of the world? Moreover, political polarization consists of within-country divergences, of which there was limited discussion in this manuscript (aside from a few aspects of the results section). The authors should either make the link between worldwide divergence in values and within country divergence more clear, or just remove some aspects of this paragraph. I think part of the confusion here stems from this paragraph seemingly contradicting the findings that within-country homogeneity was associated with greater country divergence. In this sense, the widening divergence in values on the worldwide level is somewhat in conflict within the seemingly widening divergence within countries as well.

Thank you for pointing out the lack of clarity. We agree with everything you say here. We have removed the sentences about the rise of right-wing populism because we agree it conflates the divergence of values within countries with the divergence of values between countries, and we only find support for the latter in this paper.

23. Figure 4 shows that the more homogenous a country is (regardless of emancipative or traditional values), the more distinctive they are relative to the world. But this doesn’t seem to fit with the pattern we see that Western democracies are becoming more culturally diverse over time. Even countries at the extreme end (Norway, Netherlands) are more ethnically and culturally diverse than they used to be a few decades ago.

This is an interesting point. Figure 4 does not have a temporal dimension—it just shows each country’s within-country heterogeneity at the final timepoint against the country’s value distinctiveness at the final timepoint, with colors indicating whether the country values have diverged (value distinctiveness has increased) or converged (value distinctiveness has decreased) over time.

Countries like Norway and Sweden may have become more demographically diverse over time, but they remain quite homogenous. This is reflected by their low ranking on linguistic, religious, and ethnic fractionalization. The gold standard ethnic fractionalization index was published by Alesina et al (2003) in the journal of economic growth and positioning Norway, Sweden, and the Netherlands as #37, #38, and #48 out of 190 countries on ethnic fractionalization (where country #190, Uganda, was the most ethnically fractionalized). Some less cited analyses have tried to capture temporal changes in this index (e.g., Drazanova, 2020, Journal of Open Humanities Data) and found these countries to be rising slightly in fractionalization over time, but not to the level of countries like South Africa or Spain, which are far more ethnically diverse.

24. p. 15. “Why is wealth associated with value distinctiveness in Western but not non-Western countries? One possibility is that historical events like the Protestant Reformation or the Catholic Church’s Marriage and Family polices predisposed the West to endorse emancipative values to a greater extent than other world regions.”

Again, are the authors treating “emancipative values” as the same thing as individualism (which is the theory cited here)?

Thank you again for pointing out our lack of clarity about the relationship between emancipative values and individualism. We hope that our response to point #13 has addressed this confusion.

We have also reworded that paragraph to include other potential historical events that might have contributed to Western values emphasizing tolerance and individual autonomy. We are pasted the revised paragraph below:

Our findings answer important questions about contemporary values change, but they also pose different questions. Why has rising wealth led Western, but not non-Western, nations to espouse more emancipative values? We believe that this relationship might be embedded in ecological and cultural events throughout Western history, including the advent of participatory democracy in Athens and early Rome, enlightenment and post-enlightenment philosophy, the Catholic Church’s marriage and family polices⁶¹, the French revolution, and the reformation⁶². These phenomena may have gradually solidified a Western identity focused on autonomy, primacy of individual rights over obligations to the in-group, and tolerance for breaking norms^{2,61,63}. As the world has globalized and Western nations have competed for resources on the world stage, this identity may have crystalized even further. But this does not mean that wealth or globalization should have encouraged similar values in African, East Asian, or South Asian cultures. To the contrary, growing power and

resources may prompt these countries to affirm their own traditional values, which would be consistent with our findings.

25. What about the late sociologist Ronald Inglehart's recent findings in his latest book (*Religion's Sudden Decline*), that looking at all the waves of the WVS, we see that except for the Muslim world, and possibly some segments of post-communist countries (e.g., Russia), greater existential security (that is wealth combined with widely available social safety nets) has led to a decline in religiosity. This is also happening in the United States, which for decades was resisting the worldwide trend. How are the findings reported here relate to these secularization findings?

This is a good question. We are familiar with Inglehart's book, and we believe some of the claims are very interesting and are actually consistent with our analysis, although not always in the way that Inglehart thinks.

Inglehart finds that wealth and life expectancy have been historical predictors of religious decline, and that this is especially true when using older versions of these predictors. For example, infant mortality in 1960 is better at predicting 21st century religiosity than is infant mortality in 2017 (see Chapter 6: What's Causing Secularization? Insecurity). He also finds that current-day religiosity is even more associated with "individual choice norms" than GDP per capita. Countries that promote moral values emphasizing individual choices vs. decisions that maximize fertility are those which have the lowest rates of current-day religiosity, and this explains the effect of GDP per capita. His index of individual choice norms is nearly colinear with Welzel's measure of emancipative values, because they use the same items (e.g., homosexuality/abortion/divorce is justifiable).

Inglehart explains these findings by arguing that wealth led to a rise in individualist values, which in turn led to the decline of religiosity. He explains this logic by arguing that world religions are mainly fertility mechanisms. In his words "major world religions instill pro-fertility norms, which helped societies survive when facing high infant mortality and low life expectancy" (*Chapter 1: The Shift from Pro-Fertility Norms to Individual Choice Norms*). As infant mortality has dropped and life expectancy has risen, these norms have become less important, and religion has declined.

We see merit in Inglehart's thesis. Abrahamic world religions (and many traditional religions) do promote fertility, and their pro-fertility norms have become less adaptive over history as groups have faced different population pressures. We also agree that wealth and technological decline can produce secularization, both because it brings people into contact with secular ideas (see White & Muthukrishna, 2023, <https://psyarxiv.com/8wr5d/>) and because specific technologies supplant the perceived function of religious practices (see Jackson et al., 2023, *PNAS*).

But we doubt this is the whole story. As you point out, wealth has not led to religious decline in Muslim nations such as Saudi Arabia, UAE, or Qatar. Communist countries like China also report very low religiosity, as to wealthy non-Communist countries in East Asia like Japan, and to a lesser extent, South Korea. Drops in fertility rate in Scandinavia and China have not produced renewed calls for religiosity.

We think these exceptions arise because religiosity is also an identity-defining cultural value, and religiosity has been more important for Middle Eastern countries than East Asian countries. A growing emphasis on Islam helped Middle Eastern countries distinguish their cultural identity from the secular identities of Western Europe and Russia, which both occupied Middle Eastern land in the 20th century. In the case of East Asia, supernatural beliefs have long been common (and remain common), but East Asian countries have long embraced a secular identity that dates back to Confucianism, Taoism, and samurai culture and the Heian period in Japan. As these countries have grown more wealthy and powerful, they have reaffirmed their cultural traditions: Muslim countries have become more religious, and East Asian countries have identified with more secular philosophies such as Confucianism.

We believe this perspective can also explain Inglehart's puzzling time-lagged correlation between religiosity and wealth. In the 1970s, wealth was primarily concentrated in Western Europe and Japan—highly secular world regions. By 2020, wealth was (slightly) more evenly distributed throughout the world, with large marginal gains in some highly religious Middle Eastern (e.g., Saudi Arabia), Southeast Asian (e.g., Malaysia), South American (e.g., Chile) countries. This explains why wealth was more strongly associated with religiosity in the 1970s than it is today, and this simple explanation requires no lagged causal effect of wealth on religion.

These are mostly speculations based on our hunches, and our knowledge of the data. We share them in this letter because we find the topic interesting and we are happy to reflect on it, but we think that a more serious discussion of how value divergence intersects with secularization deserves another paper entirely.

However, we do think that our findings speak to the endurance of religion in wealth Muslim countries, and we have added a paragraph to our general discussion to this point. We cite Inglehart's book in this summary, along with some other related work:

Value divergence could also explain theoretical puzzles in the social sciences. For example, there is a popular theory that rising wealth⁴ and technology⁵⁷ facilitates religious decline because it decreases existential insecurity and relieves pressure to have children⁵⁸. But this model does not explain why rising wealth is correlated with rising religiosity in many Middle Eastern countries. From a cultural identity perspective, the dynamic is less mysterious. Islam emerged as a key part of Arabic post-colonial identity throughout the 20th century, punctuated by the formation of the Muslim Brotherhood (1928), the establishment of Saudi Arabia (1932), and revolutions against Soviet and Western European governments in Iran and Afghanistan⁵⁹. Wealth and global

influence have led countries like Saudi Arabia, Qatar, and the United Arab Emirates to emphasize the values that make them distinct from the Western world.

Reviewers' Comments:

Reviewer #1:

Remarks to the Author:

I have to commend the authors for undertaking a thorough revision. One that made the contribution of this already strong paper even more evident. I am now satisfied that all concerns I raised in my previous review have been addressed. I'm particularly impressed by how robust the results are, and I appreciate the extensive additional analyses the authors conducted to address issues such as spatial autocorrelation and the changing sample of the countries included in the WVS over time. The revision also does a nice job of defining the metrics used to assess cultural divergence and heterogeneity in its various forms and is now much more accessible as a result. I also appreciate the greater attention to evolutionary accounts of cultural variation and change that is provided in the revised introduction.

I believe that this work makes an important and substantial contribution to our understanding of cultural change, and I am happy to recommend it for publication in its current form.

-Michael E. W. Varnum

Reviewer #2:

Remarks to the Author:

This version of the paper does a good job of addressing the main issues that I had with the previous manuscript. Specifically helpful are the relabeling of the variables and the additional tables, headings, explanations that make it clearer which specific measure of value distinctiveness is being used in each set of analyses. The new and revised robustness checks also do a nice job of alleviating some of my previous concerns, as well as providing even more compelling evidence of the robustness of the main results.

Overall, I think this would make a valuable contribution to several literatures in the social sciences. The introduction of this paper makes a compelling case for need more data about changing cultural values, given the many previous competing theories about how cultural values are shifting in a globalized, post-Cold War, post-colonial era. This data presents valuable evidence of how many cultures are diverging in cultural values, and which issues and countries have the greatest divergences, which suggests fruitful pathways for investigating these processes in a more localized, nuanced way in future research.

As before, the visualizations are an excellent way to convey complex information, and are a valuable compliment to the quantitative statistical results reported in the main text and supplement. My only remaining recommendation would be to possibly add a table of contents and more headings to the supplementary materials, perhaps with a bit more of a verbal summary in the supplement of what the tables/figures are intended to show, to help readers more easily navigate the vast set of supplementary results.

Reviewer #3:

Remarks to the Author:

The authors have put a lot of work into responding to the comments and criticisms of the reviewers, and the results are excellent. This is a much-improved manuscript with a clearer conceptual framework and a sharper and more comprehensive discussion. There are just a few remaining issues and questions that I have for the authors. Otherwise, this is a groundbreaking paper and an important contribution. I recommend publication.

p. 2. "Has modernization brought a global consensus on what people value as important and just?"

In my opinion, this first sentence of the abstract is an unnecessary over-reach. Many scholars consider the 19th century as the start of modernity, and some would go back to the 18th century. The time span of this study is 40 years. All we have is cultural divergence in that time period, not before. It's best to tone down this sentence or drop it altogether.

p. 6. "We found that rate of value divergence correlated with loading on the emancipative-obedient dimension, $r(38) = 0.52$, $p < 0.001$, but not the sacred-secular dimension, $r(38) = 0.11$, $p = 0.48$."

I don't entirely understand how Welzel's 2 dimensions differ, as many of the items (listed in Figure 2A) that load on the emancipative dimension should equally load on the sacred dimension. For example, isn't the item regarding attitudes toward homosexuality strongly related to religiosity and should also load on the sacred-secular dimension? Can you explain the rationale for how these two dimensions were derived? In the SI, the authors say that these two Welzel dimensions are robustly correlated. So how is that only the emancipative dimension is picking up on cultural divergence?

Relatedly, the discussion focuses on the emancipative dimension where divergence is observed, but we are left wondering about what is happening with the sacred-secular dimension. Is there no change or the opposite (convergence)? Figure 1 shows each item's relationship to the emancipative dimension but does not specify which ones reflect the sacred-secular dimension.

p. 17. "But this model does not explain why rising wealth is correlated with rising religiosity in many Middle Eastern countries."

This topic is not the main focus of the paper and I understand there's no space or the need to expand on it more than what is said already. But to my knowledge, religiosity has remained stable in the Middle East, it has not been rising (from levels that are already very high). Also, the Middle East is the only exception to the effect of rising wealth and existential security on religious decline worldwide. It's possible that the Middle East is unique and an exception to this trend because, as the authors argue, religion is more identity-defining in Muslim-majority countries (similar to how Catholicism is more identity-defining in Poland and Ireland than elsewhere). But it is also possible that this is due to cultural lag since the increases in wealth in oil-rich Middle Eastern countries happened suddenly and more recently in history. If this latter idea is correct, we should expect religious decline in the Middle East in the coming decades. Incidentally, the time lag Inglehart and others report is not that puzzling if we consider that religious decline typically happens across multiple generations (one funeral at a time).

Ara Norenzayan

Note to All Reviewers:

In addition to the revisions described in this letter, we note one other important revision: we recently found an inconsistency in the WVS data format. One of the scales in our analysis involved the qualities that respondents felt to be important for children to learn. For these items, respondents were not allowed to select more than 5 items. However, we realized that this rule was not always followed, and the rule was more likely to be followed in some countries than others. After making this discovery, we re-ran all our analyses excluding respondents ($n = 20,380$; 5% of the total sample) who indicated more than 5 important childhood qualities (violating the questionnaire format). We found that our results were the same based on whether we excluded or included these participants. We ultimately decided to exclude respondents who did not follow the questionnaire format to keep the questionnaire format consistent across countries and timepoints. We have updated our analyses accordingly, and reported our exclusion decision in the reporting summary and in our paper's methods section.

Reviewer #1 (Remarks to the Author):

I have to commend the authors for undertaking a thorough revision. One that made the contribution of this already strong paper even more evident. I am now satisfied that all concerns I raised in my previous review have been addressed. I'm particularly impressed by how robust the results are, and I appreciate the extensive additional analyses the authors conducted to address issues such as spatial autocorrelation and the changing sample of the countries included in the WVS over time. The revision also does a nice job of defining the the metrics used to assess cultural divergence and heterogeneity in its various forms and is now much more accessible as a result. I also appreciate the greater attention to evolutionary accounts of cultural variation and change that is provided in the revised introduction.

I believe that this work makes an important and substantial contribution to our understanding of cultural change, and I am happy to recommend it for publication in its current form.

Thank you for the kind words about our paper, and your helpful suggestions from the last review that helped us improve the manuscript!

Reviewer #2 (Remarks to the Author):

This version of the paper does a good job of addressing the main issues that I had with the previous manuscript. Specifically helpful are the relabeling of the variables and the additional tables, headings, explanations that make it clearer which specific measure of value distinctiveness is being used in each set of analyses. The new and revised robustness checks also do a nice job of

alleviating some of my previous concerns, as well as providing even more compelling evidence of the robustness of the main results.

Overall, I think this would make a valuable contribution to several literatures in the social sciences. The introduction of this paper makes a compelling case for need more data about changing cultural values, given the many previous competing theories about how cultural values are shifting in a globalized, post-Cold War, post-colonial era. This data presents valuable evidence of how many cultures are diverging in cultural values, and which issues and countries have the greatest divergences, which suggests fruitful pathways for investigating these processes in a more localized, nuances way in future research.

As before, the visualizations are an excellent way to convey complex information, and are a valuable compliment to the quantitative statistical results reported in the main text and supplement. My only remaining recommendation would be to possibly add a table of contents and more headings to the supplementary materials, perhaps with a bit more of a verbal summary in the supplement of what the tables/figures are intended to show, to help readers more easily navigate the vast set of supplementary results.

Thank you for your support and helpful feedback throughout the review process!

We have taken your suggestion to add a table of contents to the supplementary materials. We agree that this makes things much easier to parse.

We also appreciate your suggestion about adding a verbal summary of the figures and tables to the supplement. This was our approach in the original submission. However, Nature's formatting guidelines mandate that we present the supplementary figures and tables separately for their accompanying verbal descriptions. We do have verbal descriptions of the figures and tables in our supplementary material. However, they are in the "Supplementary Methods" section. For example, the verbal description for Supplementary Table 17 is in the section "Full Statistics for Clustering Regressions" in the Supplementary Methods.

Reviewer #3 (Remarks to the Author):

The authors have put a lot of work into responding to the comments and criticisms of the reviewers, and the results are excellent. This is a much-improved manuscript with a clearer conceptual framework and a sharper and more comprehensive discussion. There are just a few remaining issues and questions that I have for the authors. Otherwise, this is a groundbreaking paper and an important contribution. I recommend publication.

Thank you for the kind words and feedback throughout this process!

p. 2. “Has modernization brought a global consensus on what people value as important and just?”

In my opinion, this first sentence of the abstract is an unnecessary over-reach. Many scholars consider the 19th century as the start of modernity, and some would go back to the 18th century. The time span of this study is 40 years. All we have is cultural divergence in that time period, not before. It’s best to tone down this sentence or drop it altogether.

This is a fair point. We don’t mean to misrepresent the time-scale of the study. We have modified the sentence now to read *“have the last 40 years brought more global consensus on what people value as important and just?”*

p. 6. “We found that rate of value divergence correlated with loading on the emancipative-obedient dimension, $r(38) = 0.52$, $p < 0.001$, but not the sacred-secular dimension, $r(38) = 0.11$, $p = 0.48$.”

I don’t entirely understand how Welzel’s 2 dimensions differ, as many of the items (listed in Figure 2A) that load on the emancipative dimension should equally load on the sacred dimension. For example, isn’t the item regarding attitudes toward homosexuality strongly related to religiosity and should also load on the sacred-secular dimension? Can you explain the rationale for how these two dimensions were derived? In the SI, the authors say that these two Welzel dimensions are robustly correlated. So how is that only the emancipative dimension is picking up on cultural divergence?

Relatedly, the discussion focuses on the emancipative dimension where divergence is observed, but we are left wondering about what is happening with the sacred-secular dimension. Is there no change or the opposite (convergence)? Figure 1 shows each item’s relationship to the emancipative dimension but does not specify which ones reflect the sacred-secular dimension.

Thanks for pointing out the greater need for clarity here. From our reading of Welzel, the best way to describe the difference between these dimensions is that secular-sacred values are those which generally emphasize or eschew tradition, religion, and morality whereas the emancipative values are those which more specifically promote or constrain the freedom of the individual from the group.

Empirically speaking, we find that a broad set of items loaded onto Welzel’s secular-sacred dimension. Some of these items specifically mentioned the importance of religion (e.g., confidence in churches, importance of childhood religiosity). Other items cited general ethical violations (e.g., whether it is justifiable to cheat on taxes or not to pay public transit fares), and others cite more non-normative lifestyles (e.g., whether it is

justifiable to get a divorce or be gay). Emancipative values have a comparatively narrow set of loadings. They contain all the items about non-normative lifestyles—you are right that items relating to the justifiability of homosexuality load onto both sacred-secular (0.34) and emancipative-obedient dimensions (0.64)—but fewer of the items about general ethical violations or religiosity.

To make this dynamic clearer, we have modified our paragraph explaining the two dimensions:

Have certain kinds of values diverged more than others? To test this question, we measured how much each item related to Welzel's dimensions of "sacred vs. secular" values and "emancipative vs. obedient" values^{8,27}. Welzel developed these dimensions to distinguish between values which generally emphasize or eschew tradition, religion, and morality (sacred-secular values) and values which specifically promote or constrain the freedom of the individual from the group (emancipative-obedient values)¹. We found that a broad set of items correlated loaded on the secular-sacred dimension, ranging from the justifiability of cheating on taxes (0.46), to the confidence in churches (0.46), to the justifiability of euthanasia (0.37). A comparatively narrower set of items loaded on the emancipative-obedient dimension. Of the three examples above, on the justifiability of euthanasia loaded above 0.35 (0.44).

The secular-sacred dimension is not picking up on value divergence because it contains items across the whole spectrum of convergence-divergence. The secular-sacred items relating to non-normative lifestyles—which overlap with the emancipative dimension—are diverging, but there are also secular-sacred items which are showing consistent value variation over time (e.g., "Confidence: Churches") and secular-sacred were converging ("Justifiable: Cheating on Taxes"). This is why the correlation between value divergence and secular-sacred loading is lower than for the emancipative dimension, where nearly all the highest-loading items are diverging.

We have also added Supplementary Figure 1, which reproduces Figure 1 but with shading based on the secular-sacred dimension instead of the emancipative-obedience dimension. We have also pasted that figure below. We reference the supplementary figure in the main text where we present Figure 1.

Supplementary Figure 1. Value divergence across all 40 items in our analysis, as indexed by the correlation between time and value variation for each item (using a median split normalization approach). The y-axis provides each item label associated with the longitudinal WVS dataframe. The fill color indicates the absolute value of the correlation of each item with Welzel’s index of secular vs. sacred values⁸; brighter bars correlate either highly positively or negatively on this index.

We hope that these revisions have made the difference between the indices clearer.

p. 17. “But this model does not explain why rising wealth is correlated with rising religiosity in many Middle Eastern countries.”

This topic is not the main focus of the paper and I understand there’s no space or the need to expand on it more than what is said already. But to my knowledge, religiosity has remained stable in the Middle East, it has not been rising (from levels that are already very high). Also, the Middle East is the only exception to the effect of rising wealth and existential security on religious decline worldwide. It’s possible that the Middle East is unique and an exception to this trend because, as the authors argue, religion is more identity-defining in Muslim-majority countries (similar to how Catholicism is more identity-defining in Poland and Ireland than elsewhere). But it is also possible that this is due to cultural lag since the increases in wealth in oil-rich Middle Eastern countries happened suddenly and more recently in history. If this latter idea is correct, we should expect religious decline in the Middle East in the coming decades. Incidentally, the time lag Inglehart and others report is not that puzzling if we consider that

religious decline typically happens across multiple generations (one funeral at a time).

Thank you for the insight. The possibility that you raise is certainly plausible. Wealthy Arab nations may secularize after remaining wealthy for a longer time, especially with generational change. When we spoke about “rising religiosity,” we were referring to trends in the Gallup World Poll which have found rising religiosity in several Middle Eastern countries (e.g., Qatar, Kuwait) over the last 15 years (see Figure 1 in Jackson et al, 2023, PNAS), but you are also right that many other Middle Eastern countries have maintained similar and high levels of religiosity over recent decades.

Given that we can't rule on this question without looking into the future, and the question's scope goes outside of our study, we have chosen to remain agnostic. We cite both the “identity” mechanism and the “lagged secularization” mechanism. Here is our revised paragraph:

Value divergence could also explain theoretical puzzles in the social sciences. For example, there is a popular theory that rising wealth⁴ and technology⁵⁸ facilitate religious decline because they decrease existential insecurity and relieve the economic pressure to have children⁵⁹. But this model does not explain why rising wealth has not brought religious declines in Middle Eastern countries and has even correlated with rising religiosity in some of these countries. One possibility is that these declines will happen with time, especially with generational change. In other words, countries like Saudi Arabia and Qatar may not have been wealthy long enough to experience secularization. However, another possibility is that Islam has emerged as a key part of Arabic post-colonial identity. Rising wealth and influence could have led Arab countries to emphasize the religious part of their identity to distinguish themselves from the West.

If you still feel that the paragraph subtracts rather than adds from the paper, we can remove it entirely. But as it stands, we feel that the paragraph makes a contribution to the paper.

Reviewers' Comments:

Reviewer #1:

Remarks to the Author:

I recommended publication based on the last round of revision. I continue to recommend publication of this interesting and important work.

-Michael E. W. Varnum

Reviewer #3:

Remarks to the Author:

The authors have done a superb job revising the manuscript while addressing all remaining comments and issues. I expect This important paper to be highly impactful in the coming years.